

# Late Pliocene Cordilleran Ice Sheet development with warm Northeast Pacific sea surface temperatures

Maria Luisa Sánchez-Montes[1], Erin L. McClymont[1], Jeremy M. Lloyd[1], Juliane Müller[2], Ellen A. Cowan[3] and Coralie Zorzi[4].

[1] Geography Department, Durham University, Durham, DH1 3LE, UK.
[2] Alfred Wegener Institute, Helmholtz Centre for Polar and Marine Research, Bremerhaven, 27568, Germany.
[3] Department of Geological and Environmental Sciences, Appalachian State University, North Carolina, 28608, USA.
[4] GEOTOP, Université du Québec à Montréal, Montreal, H3C 3P8, Canada.

*Correspondence to*: Maria Luisa Sánchez Montes (m.l.sanchez-montes@durham.ac.uk)

**Abstract.** The initiation and evolution of the Cordilleran Ice Sheet is relatively poorly constrained. International Ocean Discovery Program (IODP) Expedition 341 recovered marine sediments at Site U1417 in the Gulf of Alaska (GOA). Here we present alkenone-derived sea surface temperature (SST) analyses alongside ice rafted debris (IRD), terrigenous and marine organic matter inputs to the GOA through the late Pliocene and early Pleistocene. The first IRD contribution from tidewater glaciers in southwest Alaska is recorded at 2.9 Ma, indicating that the Cordilleran ice sheet extent increased in the late Pliocene. A higher occurrence of IRD and higher sedimentation rates in the GOA during the early Pleistocene, at 2.5 Ma, occur in synchrony with SSTs warming on the order of 1°C relative to the Pliocene. All records show a high degree of variability in the early Pleistocene, indicating highly efficient ocean-climate-ice interactions through warm SST-ocean evaporation-orographic precipitation-ice growth mechanisms. A climatic shift towards ocean circulation in the subarctic Pacific similar to the pattern observed during negative Pacific Decadal Oscillation (PDO) conditions today appears to be a necessary pre-requisite to develop the Cordilleran glaciation and increase moisture supply to the subarctic Pacific. The drop in atmospheric $CO_2$ concentrations since 2.8 Ma is suggested as one of the main forcing mechanisms driving the Cordilleran glaciation.

## 1 Introduction

During the Neogene, the global climate transitioned from relatively warm to cooler conditions that enabled the development of ice masses in both hemispheres (Zachos *et al.*, 2001a). The Mid-Pliocene Warm Period (MPWP, 3.3-3.0 Ma) interrupts this cooling trend, with global temperatures around 2-3 °C above pre-industrial levels (Jansen *et al.*, 2007; Haywood *et al.*, 2004), and more intense warming at higher latitudes (Haywood *et al.*, 2013; Dolan *et al.*, 2015). The MPWP has been suggested as a potential analogue for the 21st century climate due to the atmospheric $CO_2$ concentrations (400 ppmv) and largely equivalent continental configurations relative to the present (Salzmann *et al.*, 2011; Raymo *et al.*, 1996, Jansen *et al.*, 2007).

Overall, the mid-Pliocene ice masses were smaller than today (Dolan *et al.*, 2011). However, the marine isotope stage (MIS) M2 (~3.3-3.26 Ma) event is characterised by a dramatic cooling in the Atlantic Ocean and is considered to be an unsuccessful attempt at a glaciation (De Schepper *et al.*, 2013). The later onset (oNHG) or intensification (iNHG) of the Northern



Hemisphere Glaciation is marked by the expansion of the Laurentide, Greenland and Scandinavian ice sheets around 2.5 Ma indicated by ice rafted debris (IRD) records from the North Atlantic Ocean (i.e. Shackleton *et al*., 1984) and the advance of the Cordilleran Ice Sheet at 2.7 Ma inferred from magnetic susceptibility measurements at ODP Site 882 in the northwest Pacific Ocean (Haug *et al*., 1999; Haug *et al*., 2005). It is still debated whether climatic or tectonic forcing was the main driver

of the North Hemisphere Glaciation (NHG) (Haug *et al*., 2005), as it cannot be explained solely by changes in isolation (Lunt *et al*., 2008). The decrease in atmospheric $CO_2$ concentrations and radiative forcing at 2.8 Ma has been identified as a potential mechanism for climate cooling of the oNHG (Seki *et al*., 2010; Martínez-Botí *et al*., 2015). However, the timing of the oNHG varies between locations based on IRD delivery, and at some locations the NHG has been set as far back as 3.5 Ma (Nordic Seas, Mudelsee and Raymo, 2005). Alternative proposals for the oNHG suggest that orogenic changes could have led to an

increase in heat transport to the North Atlantic region during the Pliocene potentially increasing precipitation in higher latitudes promoting glacial development during the Plio-Pleistocene transition (Sarnthein *et al*, 2013; Haug *et al*., 2005; Bringham-Grette *et al.,* 2013; Fedorov *et al.,* 2013; Lawrence *et al.,* 2010).

It remains unclear whether the Cordilleran Ice Sheet of North America expanded across the oNHG, although enhanced delivery of terrigenous sediments to the Gulf of Alaska (GOA; Northeast Pacific Ocean) since 2.7 Ma has been interpreted as evidence

for ice sheet growth (Gulick *et al*., 2014). The sediments of the Gulf of Alaska (GOA) record Cordilleran glaciation in the St. Elias Mountains, at present the highest coastal mountain range in the world (Enkelmann *et al*., 2015). It has been proposed that the uplift of the St. Elias Range from early Pliocene to early Pleistocene led to an increase in orographic precipitation and subsequent increase in sedimentation rates in the GOA (Enkelmann *et al*., 2015). Mountain glaciation may have developed in the St Elias mountains as early as 5.5 Ma (Reece *et al*., 2011), ultimately developing tidewater glaciers, with the high erosion

pathway shifting to the southern St Elias Range at 2.6 Ma (Enkelmann *et al*., 2015). Rather than a tectonic control on Cordilleran glaciation, an alternative explanation could be the reduced radiative forcing and climate cooling associated with the decline in $CO_2$ at 2.8 Ma. However, it is difficult to resolve these hypotheses in the absence of high resolution data for both ice sheet extent and climate from the GOA. Despite the global drop in atmospheric $CO_2$ at 2.8 Ma, it remains unclear whether the Cordilleran Ice Sheet also expanded.

Here, we present a new multiproxy data set obtained from IODP core site U1417 (56° 57.58' N, 147° 6.58' W, water depth 4218 m; Fig. 1) in the GOA. The core site allows examination of the land-ocean interactions associated with advance and retreat phases of the Cordilleran Ice Sheet across the Pliocene-Pleistocene transition, in the context of mountain uplift. The sediments were collected during IODP Expedition 341 (Jaeger *et al.,* 2014) and were analysed to reconstruct sea surface conditions by means of alkenone and IRD data covering the time interval from 4 to 1.7 Ma years ago. Terrestrial organic matter

input to Site U1417 is assessed through the abundance of long-chain *n*-alkanes and palynological analysis.

## 2 Study area

### 2.1 The Gulf of Alaska (GOA)



The GOA extends from the Alaska Peninsula in the west to the Alexander Archipelago in the east (Hogan, 2013), delimited by the Bering Sea on the west and the Alaska coast in the north and east, which is, in turn, bounded to the north by the Pacific Mountain System (Molnia, 2008). The south of the GOA connects with the North Pacific Ocean (Fig. 1). Glaciers cover 20% of the Gulf of Alaska watershed (Spies, 2007), and the major rivers draining the St. Elias and Chugach mountains towards the

GOA (the Alsek River and the Copper River), are fed by meltwater discharge which peaks in August (Weingartner, 2007). The GOA mean annual freshwater discharge derives from high precipitation, runoff and snow melt from watersheds along the SE Alaskan coast (Spies, 2007). High precipitation is due in part to the proximity of the North Pacific Ocean, as a source of moisture, and the high topography of the Pacific Mountain System driving orographic precipitation.

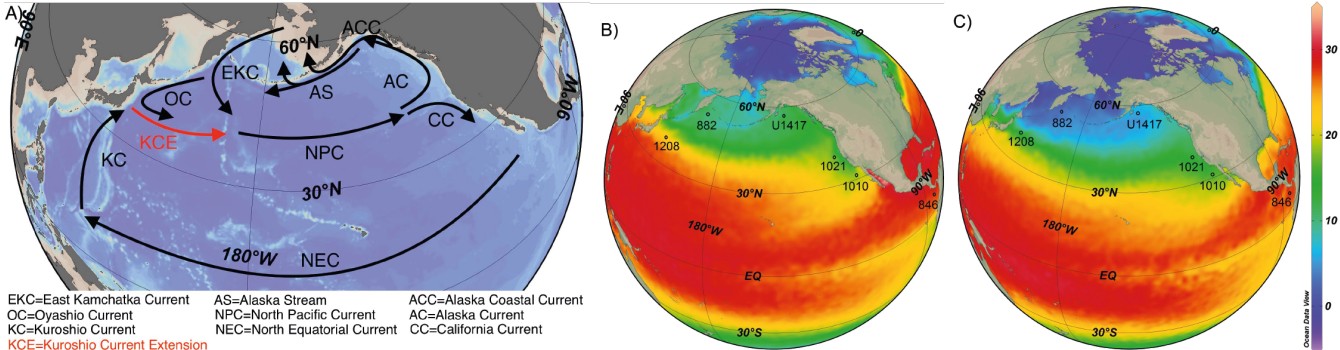

**Figure 1: Map of modern ocean circulation and SSTs.** a) Modern North Pacific Ocean circulation, b) September c) and December 1955-2013 SST average centred in the North Pacific Ocean (NOAA WOA13, Locarnini *et al*., 2013) and core sites discussed in this study. Map made using Ocean Data View (Schlitzer, 2018).

The Alaskan Coastal Current (ACC) flows anti-clockwise along the GOA coastline and westward to the Bering Sea (Fig. 1a), and its properties are dominated by nutrient and meltwater supply from the coastal Alaskan glaciers (Spies, 2007). Further
offshore, the Alaska Current (AC) also flows anti-clockwise, controlled in strength by the Alaska Gyre (Kato *et al*., 2016) (Fig. 1a). The Alaskan Gyre is, in turn, influenced by atmospheric circulation via the Aleutian Low (AL) and the Pacific High Pressure Systems, which are coupled in an annual cycle. High pressures dominate during the summer season and low pressures dominate during autumn to spring (Hogan, 2013), when the AL also migrates eastward across the North Pacific Ocean, becoming most intense when located in the GOA during winter (Pickart *et al.*, 2009). The coast of Alaska receives high winter
precipitation because of the AL winter position and strength (Rodinov *et al.,* 2007) and Alaska's high topography which drives orographic precipitation. The GOA locally receives annual precipitation of ~800 cm (Powell and Molnia, 1989). During summer, the AL is less intense and almost disappears when it is located in the Bering Sea. A weaker AL is translated into reduced precipitation over the GOA.

A strong winter AL also creates a strong zonal SST gradient in the North Pacific Ocean (Fig. 1b). During winter, the ocean
responds to a more intense AL through southward movement of the cold Arctic waters, and northward flow of mid-latitude warm waters into the Gulf of Alaska through the AC. During the summer migration of the AL northwards, the GOA registers higher SSTs due to higher insolation on the North Pacific Ocean, and as the zonal SST gradient is reduced, the storms diminish (Pickart *et al.,* 2009) (Fig. 1c).



## 3 Material and Methods

### 3.1 Age model and sedimentation rates

The shipboard age model was calculated using magnetostratigraphy (Jaeger *et al*., 2014, Fig. S1-3). The recovery of the Pliocene-early Pleistocene sediments averaged 70 % (Expedition 341 Scientists, 2014), with a number of core breaks in the record. Poor carbonate preservation across the Pliocene and early Pleistocene prevents production of a higher resolution stable isotope stratigraphy. The shipboard depth models place all discrete core biscuits to the upper depth range of each core, and a continuous core break below; it is possible that the biscuits were originally distributed through the core barrel before recovery on the ship. We have converted the depth scale of our data sets to assume an even distribution of core biscuits and core breaks (Fig. S1), converted these depths to age and interpolating the ages of the samples between core top and bottom (Fig. 2 and Figs. S1-S3). The magnetostratigraphy ages were similar between the shipboard and new age model; The Gauss/Matuyama magnetic reversal (2.581 ±0.02 Ma and 330.76 ±1 m CCSF-A) was well constrained in multiple holes to provide an important age control point for this study (Fig. S1). The shipboard age model sedimentation rates show a marked but temporary increase between 2.5-2.0 Ma, which has been attributed to the first major erosion of the landscape by expansion of the Cordilleran Ice Sheet (Gulick *et al*., 2014). Our new sedimentation rates detail a two-step increase from 2.5-2.4 and from 2.4-2.0 Ma (Fig. S3).

### 3.2 Biomarkers

A total of 119 biomarker samples between 4 and 1.7 Ma were analysed for biomarkers, which corresponds to an average sampling resolution of 19 kyr. Microwave lipid biomarker extraction was carried out following the method of Kornilova and Rosell-Melé (2003). The total lipid extract was separated into 4 fractions by silica column chromatography, through sequential elution with Hexane (3 ml), Hexane: Dichloromethane (9:1) (1.5 ml), Dichloromethane (5.5 ml) and Ethylacetate:Hexane (20:80) (4 columns) to generate: *n*-alkanes, aromatics, ketone and polar fractions.

The *n*-alkane fraction was analysed by different sets of gas chromatography (GC) configurations for compound quantification and identification. A Thermo Scientific Trace 1310 gas chromatograph was fitted with flame ionization detector (GC-FID) and a split-splitless injector. Compressed air is set as the air flow, helium (He) is set as the carrier flow, nitrogen (N) as a make-up flow and hydrogen (H) helps with ignition. The oven temperature was set at 70 °C for 2 min, then increased to 170 °C at 12 °C min$^{-1}$, then increased to 310 °C at 6.0 °C min$^{-1}$, then held at 310 °C for 35 min. *N*-alkanes were separated using a 60 m x 0.25 mm i.d., Restek RXi-5ms column. (0.25 μm 5% diphenyl-95% dimethyl polysiloxane coating). Compound identification was confirmed using a Thermo Scientific Trace 1310 gas chromatography mass spectrometer (GC-MS), equipped with a programmable temperature vaporizer (PTV) injector. He was used as a carrier flow. The oven temperature program was set at 60 °C during 2 min and then raised at 12 °C min$^{-1}$ until reaching 150 °C and then raised again to 310 °C at 6 °C min$^{-1}$ and held for 25 min. Compounds were quantified with reference to internal standards (5α-cholestane) and normalised to the original extracted dry weight of sediment, and to sedimentation rate changes by calculating the mass accumulation rate (MAR). The ratio of higher land-plant derived long-chain *n*-alkanes against aquatic sourced short-chain *n*-

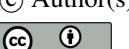



alkanes (TAR) (Eq. (1); Cranwell, 1973) and the carbon preference index (CPI) (Eq. (2); Bray and Evans, 1961) were calculated using GC-FID peak areas of the respective compounds:

$$TAR = \frac{[C27]+[C29]+[C31]}{[C15]+[C17]+[C19]} \tag{1}$$

$$CPI = \frac{\frac{[C25-33(odd)]}{[C24-32(even)]}+\frac{[C25-33(odd)]}{[C26-34(even)]}}{2} \tag{2}$$

Alkenones (ketone fractions) were quantified by a GC coupled with chemical ionisation mass spectrometry (GC-CIMS), adapted from the method of (Rosell-Melé *et al.*, 1995). Analyses were performed using a Trace Ultra gas chromatograph directly coupled to a Thermo DSQ single quadrupole mass spectrometer, fitted with a programmed temperature vaporising (PTV) injector. 1.2 ml of sample is injected. Alkenones were separated using a 30 m x 0.25 mm i.d., Restek RXi-5ms column (0.25 μm 5% diphenyl-95% dimethyl polysiloxane coating). Helium was employed as the carrier gas (2 ml min[-1]). The injector

was held at 120 °C and splitless mode (1.2 min) during injection, and then immediately temperature programmed from 120 °C to 310 °C at 10 °C s[-1], then held for 0.6 min. The oven was programmed to hold at 175 °C for 1.7 min, then increased to 310 °C at 11 min[-1], and held at from 310 °C for 12 min. The mass spectrometer was operated in positive chemical ionisation mode (PICI), using high-purity anhydrous ammonia (N6.0, BOC) introduced to the ion source through the CI gas inlet. Selected ion monitoring was performed, targeting the 8 ions corresponding to the [M + NH4][+] adducts of the target $C_{37}$ and $C_{38}$ alkenones

and the internal standard (2-nonadecanone), each with a selected ion monitoring (SIM) width of 1 m $z^{-1}$ and a dwell time of 30 min. The target m $z^{-1}$ were: 300 (2-nonadecanone), 544 ($C_{37:4}$), 546 ($C_{37:3}$), 548 ($C_{37:2}$), as detailed by (Rosell-Melé *et al.*, 1995). The alkenone $U^{K}_{37}$' index has been converted into SST according to the core-top to annual mean SST correlation constructed with samples spanning 60° S to 60° N (including from the Pacific Ocean) (Eq. (3); Müller *et al.*, 1998). The novel BAYSPLINE SST calibration (Tierney and Tingley, 2018) provides similar SST estimates in the northern latitudes than

previous calibrations. The seasonality in the alkenone production has been evidenced in the North Pacific (Tierney and Tingley, 2018). The SST calibration of Prahl *et al.* (1988) (Eq. (4)), which includes the $C_{37:4}$ alkenone, is also displayed here for comparison, as some concerns have arisen with the use of the $U^{K}_{37}$' index in samples with high $C_{37:4}$ in the Nordic Seas (Bendle *et al.*, 2005). We identify samples with high $C_{37:4}$ by presenting the percentage of $C_{37:4}$ relative to the other $C_{37}$ alkenones, %$C_{37:4}$ (Bendle and Rosell-Melé, 2004) (Eq. (5)).

$$U^{K}_{37}{}' = \frac{[C_{37:2}]}{[C_{37:2}]+[C_{37:3}]} = 0.033SST - 0.044 \tag{3}$$

$$U^{K}_{37} = \frac{[C_{37:2}]-[C_{37:4}]}{[C_{37:2}]+[C_{37:3}]+[C_{37:4}]} = 0.040SST - 0.104 \tag{4}$$

$$\%C_{37:4} = \frac{[C_{37:4}]}{[C_{37:2}]+[C_{37:3}]+[C_{37:4}]}*100 \tag{5}$$

### 3.3 IRD

IRD is quantified by weighing the coarse sand fraction (250 μm[-2] mm) following the method of Krissek (1995). Coarse sand

was separated from 10 cm[3] samples by wet sieving after air drying and rinsing with distilled water to remove salts. Each sand sample was examined with a binocular microscope to estimate the volume of terrigenous ice-rafted sediment (in volume





percent) in order to exclude biogenic components and burrow fills of manganese and pyrite, which do not have an ice-rafted origin. The mass accumulation rate of IRD (in grams per cm$^2$ kyr$^{-1}$) was calculated as in Eq. (6):

$$IRD\ MAR\ =\ CS\%\ *\ IRD\%\ *\ DBD\ *\ LSR \tag{6}$$

where CS% is the coarse sand abundance (multiplied as a decimal), IRD% is the IRD abundance in the coarse-sand fraction

(as a volume ratio), DBD is the dry bulk density of the whole sediment sample (in grams per cm$^3$) determined from discrete shipboard measurements and LSR is the interval average linear sedimentation rates (in cm kyr$^{-1}$).

Closed-form Fourier analysis was used to describe the shape of quartz grains in the IRD fraction imaged on a Quanta FEI 200 Scanning Electron Microscope (in the high vacuum mode at 20 kV) following methods that have been used to describe sedimentary particles for more than 40 years (Ehrlich and Weinberg, 1970; Ehrlich *et al*., 1980; Dowdeswell, 1982; Livsey *et*

*al.,* 2013). Two-dimensional SEM images (from 200 to 500 X magnification) were input into ImageJ to produce a line trace of the boundary for each grain. The output was inspected to verify that the trace was representative of the grain. 120 xy coordinate points were output from the boundary to represent the grain and these were input into the software program PAST (Hammer *et al.,* 2001).  Harmonic amplitudes 1-20 were calculated, lower orders (1-10) represent grain shape, a function of provenance and higher order harmonics (11-20) represent grain roundness (Dowdeswell, 1986; Haines and Mazzullo, 1988;

Livsey *et al*., 2013). An average dimensionless roughness coefficient (Rca-b) was calculated for each sample using the harmonics 16-20 for each grain in the population. Higher Rc16-20 values indicate increasing roughness and lower coefficients indicate smoother grains (Dowdeswell, 1982; Livsey *et al.,* 2013).  The roughness coefficient is calculated as in Eq. (7):

$$Rc_{a-b}\ =\ \sqrt{0.5}\ \textstyle\sum Rn^2 \tag{7}$$

Where Rn is the nth harmonic coefficient and a-b is the harmonic range used, in our case 16-20 (Ehrlich and Weinberg, 1970).

This value represents the average roundness for the grains in each sample, numbering at least 25.

### 3.4 Pollen Analysis

Palynological treatments were performed on 13 samples according to the procedure routinely used at GEOTOP (de Vernal *et al.,* 1996). Wet sample volumes were measured by water displacement and weighed after being dried. The fraction between 10 and 120 μm was treated chemically to dissolve carbonate and silicate particles with repeated cold HCl (10 %) and HF (48

%). A small drop of the final residue was mounted on a microscope slide with glycerine jelly. Before sieving and chemical treatments, one *Lycopodium clavatum* spore tablet was added in each sample to estimate palynomorph concentrations (Matthews, 1969; Mertens *et al*., 2009). Counting and identification of pollen grains and spores were carried out with a LEICA DM 5000B microscope.

## 4 Results and Discussion

### 4.1 Early and mid-Pliocene (4 to 3 Ma): early Cordilleran Ice Sheet and first glaciation attempts

Early to late-Pliocene (4 to 2.76 Ma) SSTs at Site U1417 are highly variable (max and min SST difference of 10 °C) with an average value of 8.2 °C (Fig. 2; Table 1) which is approximately 1.7 °C warmer than modern (here "modern" refers to the



averaged decadal statistical mean SST of 6.5 °C during the 1955 to 2012 time period, NOAA WOD13; Boyer *et al*., 2013). The Pliocene and Pleistocene SSTs at the GOA have a similar SST range to modern (e.g. NOAA WOD13; Boyer *et al.,* 2013; Fig. 2). The MPWP (3.2 to 3.0 Ma) contains the highest SST peak of the Pliocene, SST=12.4 °C, 5.9 °C warmer than modern SST in the GOA. The average MPWP SST of 8.9 °C is around 2.4 °C warmer than modern. Similar to the MPWP, the MG1-Gi1 period (3.6 to 3.4 Ma) warm period contains the second highest peak in SST during the Pliocene, SST=11.7 °C, 5.2 °C warmer SST than modern GOA. Other SST peaks during the MG1-Gi1 are 2-3 °C warmer than modern. The average SST during the MG1-Gi1 period is 9.5 °C, around 3 °C warmer than modern. $C_{37:4}$ concentrations during the Pliocene remain below the threshold of subpolar/subarctic water masses identified in the Nordic Seas (Bendle and Rosell-Melé, 2004) and are consistent with a warm surface ocean and/or minimal meltwater inputs to the GOA. The wide range of "warmer than modern" SSTs occurring during the MPWP together with higher than modern atmospheric $CO_2$ levels and similar continental configuration, further supports the proposal to use this time period as an analogue for future climate predictions (Hansen, 2006). The MG1-Gi1period represents the opportunity for studies to focus on a prolonged period of sustained warm SST but with similar SST peaks than the MPWP.

During the early to mid-Pliocene, IRD is absent and sedimentation rates are the lowest of the 4-1.7 Ma record. Small glaciers in Alaska since or before 4 Ma have been indicated from neodymium and lead isotope records from the Bering Sea (Horikawa *et al*., 2015). However, our data show that during the early and mid-Pliocene, the Cordilleran Ice Sheet was not yet extensive enough to erode or transport large volumes of sediment and runoff to the GOA. In contrast, IRD at ODP Site 887 (located 200 km southwest of U1417) suggests glacial influence in the GOA since 5.5 Ma (Reece *et al*., 2011). Early Pliocene and even Miocene evidence of tidewater glaciation (δ18O, IRD) has been found at other locations in the North Atlantic (Mudelsee and Raymo, 2005; Bachem *et al*., 2016). Reece *et al.* (2011) attributed the initiation of glaciation in the GOA to the uplift of the Yakutat formation. However, IRD mass accumulation rates at ODP 887 prior to 2.6 Ma are very small, being close to 0 and < 0.2 g cm$^{-2}$ Ky$^{-1}$ (Krissek, 1995). The low sedimentation rate, high TAR, low %$C_{37:4}$ and absence of IRD during this period at Site U1417 suggest that although the GOA experienced intervals of relatively cool SSTs (Fig. 2a), limited mountain glaciation but not full-scale continental glaciation resulting in tidewater glaciers marked the early and mid-Pliocene presented here.

There are two intervals of significant cooling recorded during the Pliocene at Site U1417: the MIS M2 (3.3 Ma) and KM2 (3.2 Ma) (Fig. 2a). Neither of these cold intervals record IRD delivery to Site U1417. Both intervals are punctuated by core breaks, suggesting a change in the sediment lithology which made core recovery difficult (Fig. S1). The M2 has been proposed as a significant Pliocene glaciation, though smaller than early Pleistocene glaciations, possibly due to the prevalent high atmospheric $CO_2$ levels (De Schepper *et al*., 2013) (Fig. 3a). However, if this event, and the climatic conditions we record in the GOA, triggered the appearance of glaciation in Alaska at all (De Schepper *et al*., 2013), our data suggests the glaciation was not intense enough to support an ice sheet with a tidewater margin that delivered icebergs to Site U1417. Our record provides evidence for relatively cold SST conditions during M2, as cold as conditions during major glacial cycles of the Pleistocene, but with no evidence for the development of a major Cordilleran Ice Sheet.

Between 4 and 3 Ma ago, we observe maximum TAR values (up to 16; Fig. 2), pointing to a higher export of terrigenous organic matter (i.e. land-plant leaf waxes) to the GOA. We assume that the warm and wet climate of the early Pliocene during





high atmospheric $CO_2$ levels potentially sustained a highly vegetated landscape in Alaska and west Canada which delivered high amounts of plant wax lipids and pollen grains into the GOA. The absence of IRD and higher pollen counts may refer to an airborne transport of the leaf wax lipids rather than an export via icebergs. The colder SST during the Pliocene could have promoted a deeper AL and dust driven transport of terrigenous organic matter may have developed. Strong winds could have

5      transported plant waxes to Site U1417 during the Pliocene, as is also observed in the North Atlantic during the NHG (Naafs *et al.*, 2012). Müller *et al.* (2018) also proposed an export of long-chain *n*-alkanes to the GOA via dust storms. We suggest that, in addition to wind transport, also coastal river discharge of terrigenous organic matter may have contributed to higher TAR values recorded at Site U1417.

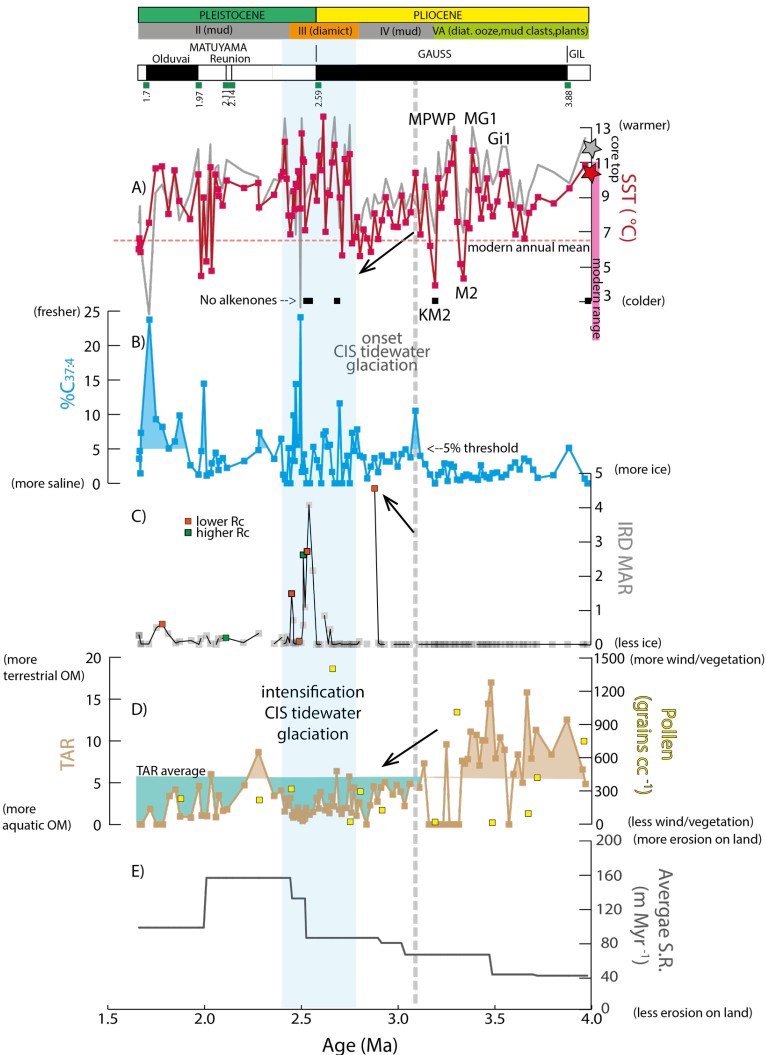

10     **Figure 2: Site U1417 across the Pliocene-Pleistocene transition.** a) red line: SST from $U^{K'}_{37}$ index according to Müller *et al*. (1998) calibration; grey line: SST from $U^K_{37}$ index according to Prahl *et al.* (1988) calibration. Black squares are samples where alkenones were not detected. Dashed red line: Modern averaged decades (1955-2012) annual statistical mean SST=6.4 °C at 0 m water depth (NOAA



WOD13, Boyer *et al.*, 2013) at Site U1417, similar to the modern annual average SST=7 °C at GAK1 station during the 1970-2018 time intervals for the 0-100 m water column depth (Weingartner *et al.* 2016) in the Gulf of Alaska; Red star on y-axis: value of our youngest sample analysed at Site U1417 (U1417D 1H-1W 44-48; 0.016 Ma; SST=10.6 °C with Müller *et al.* (1998) calibration; SST=11.8 °C with Prahl *et al.,* 1988 calibration); Pink rectangle on y axis: modern averaged decades (1955-2012) statistical mean SST during winter and summer at Site U1417 and 0 m water depth (NOAA WOD13, Boyer *et al.*, 2013) SST=0-11,3 °C; b) abundance of the cold and freshwater alkenone $C_{37:4}$ (%). Horizontal line shows the threshold of Bendle *et al.* (2005) above which subarctic/subpolar water masses were determined for the Nordic Seas; c) IRD MAR (g cm$^{-2}$ ka$^{-1}$). Orange and green squares reflect lower and higher average roughness coefficient (Rc) of the IRD quartz grains, respectively; d) terrestrial/aquatic *n*-alkane index (TAR), horizontal line shows the average TAR value, yellow squares represent pollen grains concentrations in grains cc$^{-1}$ and e) average sedimentation rates (see Fig. S3b) in m Myr$^{-1}$ from Site U1417. Upper panel: Pliocene-Pleistocene boundary, magnetostratigraphy events and interpretations (see Fig. S2 and S3) and Lithostratigraphic units of Site U1417 with simplified lithology (orange colouring represents ice rafted diamict interbedded with mud, brown colouring represents marine mud and green colouring represents diatom ooze interbedded with debris flow deposits containing mud clasts and plant fragments) (Jaeger *et al*., 2014). Vertical grey line represents the onset of the Cordilleran Ice Sheet (CIS) glaciation (or oNHG) climate transition at 3 Ma, blue shading represents the 2.5-2.0 Ma climate transition with the intensification of the Cordilleran Ice Sheet (CIS) tidewater glaciation (or iNHG) as in Table 1.

We further note that rivers and ocean currents could have transported bedrock material from the Yakutat Terrain (Childress, 2016) to Site U1417, 700 km offshore from the Alaskan coast. This would imprint the sediments delivered to the ocean with an ancient signal of terrigenous organic matter, rather than reflecting erosion of contemporary 'fresh' organic matter from vegetation and soils. The CPI is often used to estimate the maturity of the organic matter and determine its source. Previous studies suggest that elevated TAR values and CPI values close to 1 reflect coal particles found in sediments in the GOA (Rea *et al*., 1995; Gulick *et al*., 2015). However, the coal-bearing Kulthieth rocks (McCalpin *et al*., 2011), have a TAR signature of a maximum value of 2 (Childress, 2016). Since Site U1417 TAR and CPI values (> 1) do not overlap with TAR and CPI (< 1) values found onshore (Childress, 2016), we exclude a bedrock source to be driving the TAR variations at Site U1417.

### 4.2 The late Pliocene onset of the Cordilleran Ice Sheet glaciation (3 to 2.8 Ma)

The interval from 3 to 2.8 Ma is characterised by a shift of climate conditions from those observed during the early and mid-Pliocene (Fig. 2) to more glacial conditions. At 3 Ma, average SSTs at Site U1417 remain relatively warm (around 8 °C), yet, there is first evidence for a cooling at Site U1417 deduced from $C_{37:4}$ crossing the threshold of 5 % (Bendle *et al*.; 2005). %$C_{37:4}$ increases can be related to colder sea surface conditions, but at Site U1417 we suggest that increases in %$C_{37:4}$ relate to meltwater discharge from the expanding ice-sheet. From 3.1 to 2.8 Ma, SST decreases gradually from 8 to 5.5 °C (Fig. 2a) recording again colder SSTs than the modern GOA. We attribute this 0.3 Ma progressive cooling to the oNHG in response to the overall decrease in the atmospheric $CO_2$ (Seki *et al.,* 2010; Martínez-Botí *et al*., 2015). From 3 Ma, TAR values decrease to below the average of the entire TAR record, indicating that transport of leaf-wax lipids to Site U1417 slightly decreased, which may be related to a reduction in vegetational growth on land due to the advancing ice-sheet. The coincident increase in average sedimentation rates (from 65 to 79 m Myr$^{-1}$) indicates a more efficient erosive agent onshore than before 3 Ma and/or a change in the source of terrestrial matter. However, CPI values at Site U1417 remain similar to early and mid-Pliocene values, which suggests a similar source of the terrigenous organic matter. Alternatively, if the land was becoming increasingly ice covered, the growth of vegetation and the abundance of terrigenous organic matter that could be eroded and transported to the ocean would have declined, even if erosion rates as a whole increased.



The peak in $\%C_{37:4}$ at 3 Ma is followed by lower $\%C_{37:4}$ values (close to 5 %) and the first significant pulse of IRD identified by a single sample with the highest IRD MAR. This IRD MAR peak (4.5 g cm$^{-2}$ ka$^{-1}$) and an increase in sedimentation rates (from 79 to 85 m Myr$^{-1}$) at 2.9 Ma constitute the first evidence that tidewater glaciers were present in southwest Alaska delivering icebergs to Site U1417. IRD quartz grains do not appear crushed or abraded by glacial activity indicating small

tidewater valley glaciers producing icebergs which could contain grains that were introduced by rockfall or fluvial sediment. The abrupt peak in IRD delivery to U1417 at 2.9 Ma could be due to ice growth on land and cold enough SSTs to permit distal iceberg-drift and release of debris to Site U1417. The increase in sedimentation rate has been suggested to mark the maximum Cordilleran Ice Sheet extension during the Pliocene (Gulick *et al.*, 2015). Following this first peak, IRD MAR decreases to values between 0 and 1 g cm$^{-2}$ ka$^{-1}$ until 2.6 Ma. This abrupt decrease in IRD indicated lower iceberg delivery to Site U1417.

A synchronous increase of $C_{37:4}$ above 5 % suggests the melting of tidewater glaciers was responsible for the decrease in iceberg delivery despite the cold SST. Atmospheric $CO_2$ concentration peaks during this time (Fig. 3a) may have contributed to a reduced ice sheet due to radiative forcing.

### 4.3 The intensification of the Cordilleran Ice Sheet glaciation (2.7-2.4 Ma) and its evolution during the early Pleistocene (2.4-1.7 Ma)

At Site U1417, the iNHG during the Plio-Pleistocene transition (PPT) is characterised by a rise in SST, followed by highly variable values (between 5.6 to 13.6 °C) with an average of 9.7 °C, 3.2 °C warmer than modern. The iNHG is defined here as the period containing sustained signs of glaciation (i.e. Maslin *et al.*,1996; Bartoli *et al.*, 2005), which at Site U1417 are confirmed by glacial meltwater and IRD delivery. The relatively high $\%C_{37:4}$ (up to 24 %) in the early Pleistocene correlates well with the period of high IRD delivery (up to 4 g cm$^{-2}$ Ka$^{-1}$) between 2.7 to 2.4 Ma (Fig. 2b and c). This suggests this period

marks an expansion/intensification of Cordilleran glaciation following the gradual SST cooling during the oNHG.  The lithology at Site U1417 includes diamict layers that alternate with bioturbated mud from 2.7 Ma, indicating that the Cordilleran Ice Sheet remained very variable after the oNHG and maintained glacial tidewater margins discharging icebergs into the sea. Yet the intensification of the Alaskan tidewater glaciation occurred with a GOA that was overall either warmer than, or at least as warm as the mid to late Pliocene (considering Müller *et al*. 1998 SST calibration error of ±1.5 °C).

The overall increase in $\%C_{37:4}$ in the GOA during the early Pleistocene coincides with an SST warming (ca. 1 °C relative to the Pliocene; Fig. 2a and b), suggesting a stronger link between $C_{37:4}$ and meltwater fluxes rather than an expansion of subarctic water masses. Additionally, maxima and minima in $\%C_{37:4}$ during the iNHG are unrelated to elevated or lowered SSTs, respectively. There is no information available about the origin of $C_{37:4}$ in the North Pacific to explain the high $\%C_{37:4}$ values recorded at Site U1417, nor their association with intermediate SSTs rather than minima/maxima. It has been suggested that

stratification of the water column due to glacier discharge in the North Pacific could result in warmer sea surface in comparison to deeper water masses due to an increase in surface absorption of solar radiation (Meheust *et al*., 2013). Haug *et al*. (2005) proposed this could lead to an increase in ocean evaporation and orogenic precipitation, ultimately encouraging North American ice sheet growth.





Over the iNHG, low TAR values (< 1) and small variations in IRD MAR (the order of 0.1 to 2.8 g cm$^{-2}$ Ka$^{-1}$) coincide with intermediate SSTs (7 to 11 °C) and %C$_{37:4}$ between (2-24 %).This could point to an increase in marine productivity export related to an enhanced nutrient delivery to Site U1417 via glacial runoff. IRD peaks are typically present during SST minima suggesting the importance of SSTs in the delivery of icebergs to distal sites such as Site U1417. The average Rc of IRD is low

(Fig. 2c) even during IRD MAR peaks, indicating minimal glacial crushing during the iNHG. In comparison, samples from 1.6 - 1.5 Ma show a higher Rc and appear to have greater evidence of glacial crushing, suggesting development of a larger ice sheet or scouring and evacuation of sediment from the non-glacial, weathered landscape. This could indicate that at first smaller valley glaciers removed sediment and weathered rock from the landscape rather than eroding bedrock. The IRD Rc characterization suggests landscape changes that could explain the smaller variations in TAR during the early Pleistocene.

The comprehensive data set obtained from Site U1417 sediments (Fig. 2) supports a climate role in the ice-sheet expansion during the early Pleistocene and the iNHG, with an increase in precipitation from a warmer and/or stratified ocean, and cooler periods associated with IRD delivery. An alternative, or additional, explanation for the changing TAR is that tectonic uplift of the Chugach/St Elias area from 2.7 Ma (Enkelmann *et al*., 2015) led to orogenic precipitation and a change in erosional pathways (Enkelmann *et al*., 2015). The glaciation could have altered the main source of terrestrial input to the Surveyor

Channel, to higher metamorphic and plutonic sources with lower or null TAR values (Childress, 2016). An increase in CPI variability to 2 and 3 during the early Pleistocene support the change of source of organic matter delivered to Site U1417 away from the more mature coal bedrock. Lower pollen counts suggest a less vegetated landscape, which could help explain the overall lower TAR during the early Pleistocene in comparison with the Pliocene.

**4.4 The Pliocene and Pleistocene climate across the North Pacific Ocean.**

Average Pliocene SST values (4.0 to 2.8 Ma) at Site U1417 are 1 °C colder than the average early Pleistocene values (2.7 to 1.7 Ma). A warming trend from the late Pliocene to early Pleistocene observed at Site U1417 has also been observed at ODP Site 882 in the subarctic Pacific (Martínez-García *et al*., 2010), DSDP Site 593 in the Tasman Sea (McClymont *et al*., 2016) and Site 1090 (Martínez-García *et al*., 2010) in the South Atlantic. These warmings contrast with the gradual surface cooling from the Pliocene to present within the North Atlantic (e.g. ODP Site 982, Lawrence *et al*., 2009). The North Pacific warming

occurs despite an atmospheric CO$_2$ drop from 280-450 ppmv to 250-300 ppmv (similar to pre-industrial levels) from 3.2 to 2.8 Ma (Pagani *et al*., 2010; Seki *et al*., 2010) and an associated reduction in global radiative forcing (Foster *et al*., 2017). This warming signal in the GOA (and the north Pacific more generally) implies an important role for local or regional processes. We have discussed above the potential role played by ocean stratification in the North Pacific, and a possible link to the evolving Cordilleran Ice Sheet in the GOA through evaporation/precipitation feedbacks. The synchrony of these changes with

observed tectonic uplift (e.g. Enkelmann *et al*. 2015) makes it difficult to disentangle the potential climatic and tectonic mechanisms behind ice sheet expansion.

By comparing the Site U1417 SST records with others recovered from the North Pacific Ocean (Fig. 3), the following patterns emerge. All records, which range from the equator to the subarctic (Fig. 3), show warm SST during the MG1-Gi1 and the MPWP. All show orbital scale SST oscillations, including cooling during the M2 and KM2 cold events which interrupt late




Pliocene warmth (Fig. 3). All records show a cooling interval immediately following the MPWP (~3.0-2.7 Ma). The subsequent warming from 2.7 Ma observed at the subarctic Pacific Sites U1417 and 882 is also identified at Site 1010 and potentially at Site 1021 (mid-latitude east Pacific). In contrast, long-term cooling trends mark the early Pleistocene for the mid-latitude west Pacific (Site 1208) and tropical east Pacific (Site 846). This cooling trend is more consistent with the

5     development of a cooler and/or more glaciated climate as recorded in the benthic δ18O stack (Fig. 3) and in SST data from the Atlantic Ocean (e.g. ODP Site 982, Lawrence *et al.*, 2009). The regional scale warming into the early Pleistocene observed in the subarctic Pacific and mid-latitude east Pacific suggests that while local factors such as tectonic uplift could have been involved in the Cordilleran Ice Sheet development and might have influenced the SST in the GOA, the GOA climate signals reflect a wider scale warming in the mid to high latitude Pacific.

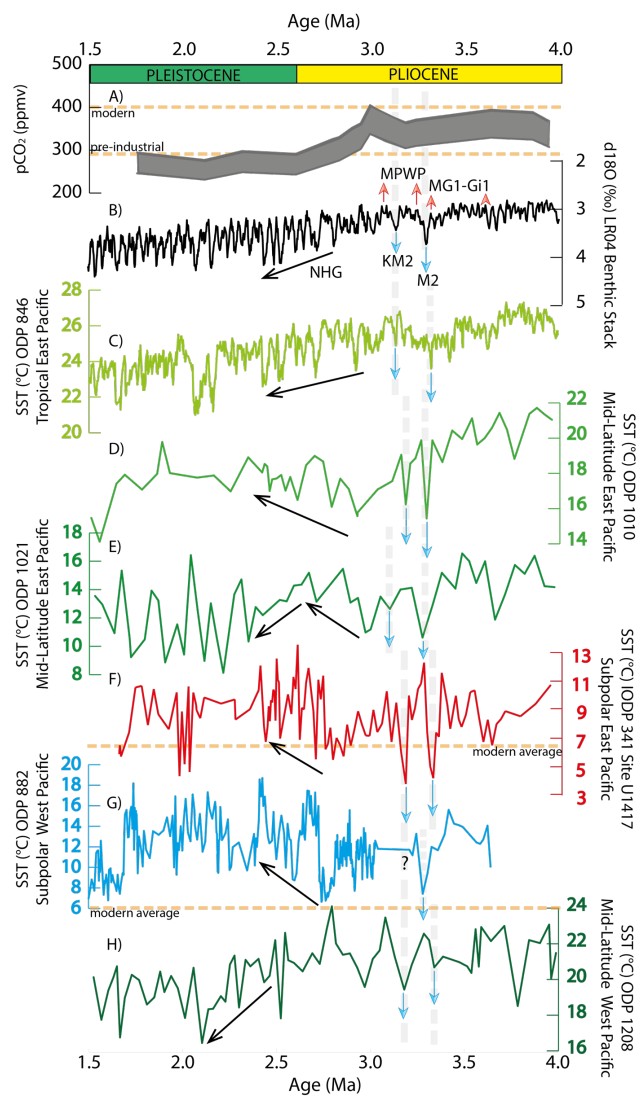

**Figure 3**: **Pliocene-Pleistocene SST across the North Pacific.** a) Alkenone pCO2 upper and lower end (ppmv) estimates at Site ODP 999A (Seki *et al.,* 2010);  b) δ18O (‰) LR04 Benthic d18O Stack (Lisiecki and Raymo, 2005); Alkenone SST (°C) from c) ODP Site 846 (Herbert





*et al.*, 2017), d) Site 1010 (Herbert *et al.,* 2018), e) Site 1021 (Herbert *et al.,* 2018), f) IODP 341 Exp. Site U1417 (Sánchez-Montes *et al.,* 2019), g) ODP Site 882 (Martínez-García *et al.,* 2010) and h) ODP Site 1208 (Herbert *et al.,* 2018). Orange horizontal lines indicate reference levels of pre-industrial times and/or modern values for each of the records or sites (SSTs from NOAA WOD13, Boyer *et al.,* 2013). Arrows indicate position of M2 and KM2 periods and cooling or warming trends across the Pacific Ocean. The KM2 event is located within the
MPWP. The MG1-Gi warming precedes the M2 event.

To understand the mechanisms that caused warmer SSTs to develop across the North Pacific under lower atmospheric $CO_2$ levels, the modern climate system is used here as an analogue to infer feasible changes in ocean and atmospheric circulation. During the Pliocene and Pleistocene, warm events are marked by a decrease in the latitudinal and longitudinal SST gradients than during cold periods (Table 3). SST gradients reduce greater on the east than the west Pacific and to a lesser decree between
east and west Pacific (Table 3). These patterns aren't consistent with modern winter/summer circulation respectively (Fig. 1), where there is a longitudinal seesaw in SST gradients: during modern winter (Fig. 1b) latitudinal SST gradients increase in the east and decrease in the west Pacific and the opposite occurs during summer (Fig. 1c). Consequently, longitudinal east-west SST gradients increase during winter and the heat budget of the west increases in comparison to the east during summer. This seems to be caused by a great seasonal SST difference at Site 1208 during modern winter, where SST cools up to 10 °C. During
the strongest AL during winter, wind and ocean currents efficiently cool the entire North Pacific, especially the west mid-latitudes. During the summer weakening of the AL and weaker atmospheric and ocean circulation, leads to warming of the North Pacific at all latitudes. During the Pliocene and Pleistocene, the latitudinal SST gradients are similar to, or increased in comparison to modern and longitudinal SST gradients are comparable or reduced. The heat exchange during the Pliocene and Pleistocene was more efficiently distributed latitudinally than at present, whereas at present it is more efficiently distributed
longitudinally across the North Pacific. During the oNHG SSTs are slightly warmer than during M2 across the mid-latitudes and subarctic sites (Fig. 3). This suggests that heat transport to the GOA is a key process in the development of the Cordilleran Ice Sheet through an increased ocean evaporation-orogenic precipitation mechanism. A key source of moisture during the Pliocene and Pleistocene could be from the stable warmth in the west mid-latitude Pacific SST.

In comparison to modern, the main characteristic of the Pliocene and Pleistocene oceanographic system is the difference in the
SST relationship between ODP 882 and Site U1417 (Fig. 3). Site U1417 shows a reversal of the modern subarctic east-west SST gradient in the Pliocene and early Pleistocene with SSTs 3-4 °C warmer in the west than in the east (Fig. 3f and g) in comparison to the modern mean annual gradient of 0.5 to 1 °C warmer in the east than west (Table 3). Additionally, there is a variation in SST gradients between warm and cold events during the Pliocene and Pleistocene concentrated across the subarctic Pacific and the east-mid-latitudes, with warmer SST in the west (ODP 882) than east (ODP 1021) during warm events and the
opposite during cold events. This is very different to the modern pattern, where ODP 882 is permanently colder than Site 1021 and Site U1417. Modern seasonal climate analogues cannot be used to explain to Pliocene and Pleistocene subarctic SST distribution. Alongside the seasonal cycle of ocean and atmospheric circulation, the AL is currently linked to the wider Pacific Ocean circulation by the Pacific Decadal Oscillation (PDO). The PDO index characterised by Pacific SST variability tendency with a periodicity of 20-30 years (Furtado *et al.*, 2011). The positive (negative) phase of the PDO is characterized by negative
(positive) SST anomalies in the central North Pacific and surrounded by positive (negative) SSTs along the North American coast and in the east equatorial Pacific. The negative phase of the PDO is a potential modern analogue for the subarctic SST



inversion during the Pliocene and Pleistocene (colder east Pacific and warmer west Pacific). The negative phase of the PDO (-PDO) is associated with an increased strength of winds off the coast of Alaska towards the northwest Pacific. Negative PDO-like conditions along with a major Cordilleran Ice Sheet over southwest Alaska could reinforce such a temperature inversion which could induce colder SSTs in the GOA (east Pacific) while producing warmer SSTs in the west Pacific.

The warmer SST characteristics of the Northwest Pacific compared to modern together with the highest SST range (Table 2, Fig. 3) and coldest SST recorded at Site U1417 (4 °C and 2 °C colder than the west subarctic Pacific) supports the hypothesis of the Cordilleran Ice Sheet meltwater influence in SST distribution across the east North Pacific during the Pliocene and Pleistocene. The comparison between Pliocene-Pleistocene and modern SST suggests that the development of the Cordilleran Ice Sheet extended the influence of cold water into the east subarctic Pacific occasionally reaching mid-latitude east North

Pacific during warm intervals turning the climate into a negative PDO-like climate. In summary, the heat contained in the west mid-latitudes to subarctic Pacific was expanded to the east during cold intervals and extended northwards during warm intervals. The -PDO-like climate was intensified during warm intervals with a decreased influence of the AL in the central North Pacific. The displacement of the AL outside the GOA allowed the interplay of secondary circulation patterns: wind transport from coastal Alaska with a south component. Weaker AL associated winds during cold intervals translated into colder

east than west subarctic Pacific, sometimes reaching mid-latitude east Pacific. A nutrient leakage of Northwest Pacific waters into the Bering Sea and via the Kamchatka Straight has been suggested after the oNHG (März *et al*. (2013); Swann *et al*. (2016)) and supports the circulation change in the subarctic Pacific during the Pliocene and Pleistocene. A reduced Arctic sea ice cover (Knies *et al.,* 2014), smaller Northern Hemisphere Ice Sheets before the onset of NHG in comparison with the present (i.e. Willeit *et al.,* 2015) and the atmospheric re-organization due to the intensification of the Asian monsoon (Rea *et al*., 1998;

Nie *et al.,* 2014) further explains a warmer west than east subarctic Pacific and atmospheric re-organization and increased moisture supply during the Pliocene and Pleistocene. In addition, our vegetation changes and climate reorganization during the Pliocene and Pleistocene agrees with palynological reconstructions from lake El'gygytgyn in Arctic Russia (Andreev *et al.,* 2014). A reversed ocean circulation has been suggested since the mid-Pliocene in the Bering Sea via modelling studies (Matthiessen *et al.,* 2008) which is supported by the restriction in Arctic water masses influencing the SST of the west subarctic

Pacific and the change in ocean circulation in the subarctic Pacific during the Pliocene and Pleistocene in comparison with modern (Fig. 3). This may have weakened or supressed the influence of Oyashio Current and greater influence of the Kuroshio extension (Fig. 1a) due to a reduction in ocean circulation characteristic of the +PDO. März *et al*. (2013) and Swann *et al*. (2016) also suggest a connected Bering Sea and west subarctic Pacific via the Kamchatka Strait with an influence of North Pacific freshwater into the Bering Strait. The presence of freshwater across the subarctic Pacific during the early Pleistocene

(Site U1417 and ODP 882; Martínez-García *et al.,* 2010) supports the studies of März *et al*. (2013) and Swann *et al*. (2016). Our data cannot confirm the presence of sea ice in the GOA but the absence of alkenones in samples next to samples recording the coldest SST at Site U1417 could suggest sea ice conditions. The new SST data presented here is the first climatic data for the GOA covering this time period (and one of only a few for the North Pacific), and may therefore be important for improving assessment of climate model outputs for the Pliocene to early Pleistocene period (Dolan *et al*., 2015).



## 5 Conclusions

The sea surface temperature (SST) evolution from the Pliocene to the Pleistocene in the subarctic Northeast and east-mid latitude North Pacific is very different from the North Atlantic, with a colder late Pliocene than early Pleistocene. The early Pliocene appears to be characterised by a weak, negative Pacific Decadal Oscillation "PDO-like" circulation where there is no obvious noticeable glaciation in the St. Elias mountains. A series of cooling events during the Pliocene (including the M2 event) could have initiated glaciation in Alaska but it was limited to mountain glaciers probably due to high atmospheric $CO_2$ concentrations, the lower topography in coastal Alaska and the inefficient supply of moisture to the high latitudes.

The overall increase in latitudinal SST gradients and stable warm west mid-latitude Pacific during the Pliocene and Pleistocene compared to modern was a key mechanism increasing moisture supply to the Gulf of Alaska (GOA) for the growth of the Cordilleran Ice Sheet. A reversal of the east-west SST gradient relative to today and similar to modern negative PDO-like climate could have been a more efficient mechanism of delivering moisture to the GOA from the west subarctic Pacific to the subarctic since the MPWP. During warm periods, the enhanced negative PDO-like climate with the associated changes in atmospheric and ocean circulation intensifies and increases the moisture transport from mid-latitude ocean evaporation to the subarctic. During cold periods, negative-PDO ocean and atmospheric circulation weakens and peaks in ice rafted debris reach Site U1417. Unlike during the Pliocene, the early Pleistocene drop in atmospheric $CO_2$ concentrations could have been decisive in developing a continuous glaciation of the Cordilleran Ice Sheet during the variable climate of the intensification of the Cordilleran tidewater glaciation. The synchronous tectonic uplift of the St Elias mountains could also have been a contributing factor for the Cordilleran Ice Sheet expansion, increasing the potential for precipitation as snow over the ice sheet source regions, despite warm SST in the GOA. Moisture supply to the GOA by an enhanced -PDO like climate is a key mechanism for glacial growth during the early Pleistocene.

## Data availability

The data presented in this manuscript has been submitted to Pangaea.de and it is under review. After the publication of this manuscript, the data would be accessible through this link  https://doi.org/10.1594/PANGAEA.899064 and could be cited as Sánchez-Montes *et al*. (2019). The SST data in this publication will also be published in the PlioVAR database.

## Author contribution

New data sets presented in this manuscript derive from the PhD project of MLSM supervised by ELM and JML. JM was closely engaged from early stages of this project including aspects of method development. EAC generated the IRD data, and CZ generated pollen data. All authors have contributed to data interpretations. MLSM prepared the manuscript with contributions from all co-authors.



**Acknowledgements**

We would like to acknowledge the International Ocean Discovery Program U.S. Implementing Organization (IODP-USIO) and the captain and crew of the D/V *Joides Resolution*. This work was supported by funding from Van Mildert College and the Durham Doctorate Scholarship (MLSM), the Philip Leverhulme Prize (ELM), and a NERC-IODP grant (NE/L002426/1)

to ELM. JM received funding through the German Research Foundation (MU3670/1-2), an ECORD grant and the Helmholtz Association (VH-NG-1101).  EAC received funding from U.S. National Science Foundation award OCE-1434945 and a post-expedition award from the U.S. Science Support Program of the IODP. We thank valuable comments from Professor Joseph S. Stoner on our new age model proposed here. We thank Martin West, Amanda Hayton and Kathryn Melvin for assistance with the GCMS analyses, and Mathew Sandefur and Hugh Harper for assistance with the IRD analyses. This work is a

contribution to the PlioVAR working group synthesis of Pliocene marine data (http://pastglobalchanges.org/).

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





**Table 1: Average SST (°C) and %C$_{37:4}$ during key climatic intervals 4.0-3.0 Ma, 3.0-2.8 Ma, 2.7-2.4 Ma, 2.4-1.7 Ma.** Average SST (°C) is the average of all the data points of the time interval, peak SST (°C) average is the average of the highest data points of each interval selected (Fig. 2), trough SST (°C) average is the average of the lowest data points of each interval (Fig. 5.2) and the average SST (°C) variability is the difference between average SST peak and the average SST trough. In black: data calculated from U$^K_{37}$ (Prahl *et al*., 1988) and in black bold, data from U$^K_{37}$' (Müller *et al*., 1998).

| Age intervals (Ma) | Average SST (°C) | Peak SST (°C) average | Trough SST (°C) average | Average SST variability (°C) | Average C$_{37:4}$ (%) | Peak C$_{37:4}$ (%) |
|---|---|---|---|---|---|---|
| 4.0-3.1 | 10.2/**8.7** | 12.5/**11.4** | 7.2/**4.4** | 5.3/**7.0** | 1.9 | 10.5 |
| 3.1-2.8 | 8.5/**7.3** | 9.9/**9.0** | 7.4/**5.7** | 2.4/**3.3** | 3.9 | 4.9 |
| 2.7-2.4 | 10.2/**9.8** | 13.2**12.6** | 8.4/**6.6** | 4.8/**5.9** | 4.8 | 24.1 |
| 2.4-1.7 | 9.0/**8.6** | 10.7/**10.4** | 6.8/**4.8** | 3.9/**5.6** | 5.2 | 23.8 |
| 4-2.8 Ma | 9.6/**8.2** | 11.2/**10.2** | 7.3/**5.1** | 3.9/**5.2** | 2.5 | 10.5 |
| 2.7-1.7Ma | 9.6/**9.1** | 12.0/**11.5** | 7.6/**5.7** | 4.4/**5.8** | 5.0 | 24.1 |

**Table 2: Extent of SST range and influence across core sites of Fig. 3.** SST cooling or warming at significant climate intervals: the cooling Pliocene prior the M2 event, the MG1-Gi1 warming, the M2 cooling event, the onset of the NHG, the intensification of the NHG and the period after the NHG during the early Pleistocene. Cells highlighted in red and blue shading are highest SST range (darker red or blue) to lesser degree of SST range (lighter red or blue) of each interval.

| Time intervals | ODP 846 | ODP 1010 | ODP 1021 | Site U1417 | ODP 882 | ODP 1208 |
|---|---|---|---|---|---|---|
| *Plio* | *3°C* | *3°C* | *4.5* | *4°C* | *?* | *3°C* |
| *MG1-Gi1* | *2.5°C* | *2°C* | *2°C* | *4°C* | *4°C* | *3°C* |
| *M2* | *2°C* | *5°C* | *5°C* | *8°C* | *5°C* | *1°C* |
| *oNHG* | *2°C* | *3.5 °C* | *2.5 °C* | *5°C\** | *9°C\** | *4°C* |
| *iNHG* | *4°C* | *3°C* | *5°C* | *8°C* | *10°C* | *6°C* |
| *ePleist* | *4°C* | *6°C* | *8°C* | *8°C* | *11°C* | *6°C* |
| *ePleistMAX* | *4°C* | *?* | *7°C\** | *7°C* | *8°C\** | *4°C* |

*(\*) SST cooling to 6 °C during the oNHG.*

**Table 3: Average SST gradients across the North Pacific during cold and warm Pliocene and Pleistocene intervals and during modern summer and winter.** Calculated from Fig. 3 SSTs and Fig.1b and c based on intervals at Table 2.

| Period | Latitudinal SST average gradients | | Longitudinal SST average gradients |
|---|---|---|---|
| | East (ODP 846 - U1417) | West (ODP 1208 - ODP 882) | East-West |
| Cold | 17.8°C (±1.5°C) | 13.8°C (±2°C) | 3.9°C (±1°C) |
| Warm | 14.5°C (±1.5°C) | 10.8°C (±0.5°C) | 3.1°C (±0.6°C) |
| Winter | 12-14°C | 6°C | 8-6°C |
| Summer | 8°C | 12°C | -4°C |