# Peer review of "Late Pliocene Cordilleran Ice Sheet development with warm Northeast Pacific sea surface temperatures"

_Climate of the Past, 2019_

## Short Comment (SC1) · 14 Mar 2019

Sánchez-Montes et al. present new and exciting SST and IRD data from IODP Site U1417 in the Gulf of Alaska spanning 4 to 1.7 Ma. This occurs during the Pliocene to Pleistocene transition when the world shifted into the modern ice age period. By measuring IRD and SST records in the same core, the authors can provide a one-to-one comparison between local temperature and the behavior of tidewater glaciers.

While this data is exciting, the paper needs some substantial revisions and consideration of other data not referenced. I'll leave my comments at a more general level at present to guide revision prior to more specific comments.

First, don't call it the mid-Pliocene warm period. The MPWP is actually the mid-Piacenzian warm period. The Piacenzian used to be in the mid Pliocene but as of 2009, it is now the late Pliocene. See Gibbards et al.'s revision of the Cenozoic timescale.

Second, the authors should skip this comparison with modern SST in their discussion and rather focus on comparing with the alkenone record of Praetorius et al. (2015 Nature) that is from 17 ka-0 ka at what is now Site U1419 (core is EW0408-85JC taken on the site survey cruise for the IODP leg). The core top temp is the same as at U1417, supporting the comparison. And more importantly, the glacial interglacial absolute and relative change at U1419 from 17 to 0 ka is the same as the Pliocene and early Pleistocene range and absolute temps at U1417. I think this rules out a major change in $CO_2$ average composition as a driver of a Pliocene to Pleistocene to last deglaciation SST pattern.

Third, quit saying this is the Cordilleran ice sheet. IRD only means you have a marine terminating ice margin. There is no Cordilleran ice sheet today and the Gulf of Alaska has a lot of marine terminating ice margins and icebergs floating around. For instance, both Bering and Malaspina glaciers could quickly become marine terminating if a big storm came through and blew away their morainal banks that happen to right now be above sea level. Both glaciers have beds below sea level except for that little bank. The evidence for an ice sheet comes from the dated proglacial gravels to the northeast in Hidy et al. (QSR 2013) that date the first Cordilleran ice sheet to about ~2.6 Ma. This paper should be discussed. Likewise, the authors should use the proximal mag sus. record from ODP Site 887 rather than the far removed to western Pacific records of 882 that Haug et al. produced. I would also include comparison of the U1419 mag sun record to 887 to support the authors suggestions/conclusions. The mag sus record could also help in improving the IRD resolution/interpretation. Surprised it isn't included. In summary, the authors should just refer to tidewater glaciation of the mountains, leaving out the word ice sheet or Cordilleran ice sheet. As far as the record they have in U1419, the conclusion is that some icebergs survived to U1419 once at ~2.7

Ma and then again after ∼2.6 Ma. This is important findings but by no means says anything about an ice sheet or its size. The authors could compare IRD abundance to the IRD record from Addison et al. (2012 Paleoc) on 85JC. Now, 85JC is much closer to the ice margin and coast but could provide some kind of context.

Fourth, I would greatly reduce to just cut the discussion of the PDO or analogues to modern SST patterns from the paper. The whole section is very confusing and hard to follow. Likewise, this depends highly on the age models of all the cores and these are not discussed. To make such comparisons/conclusions, common age models and uncertainties need to be applied which I think is beyond the scope of this paper. Rather, the authors should smooth down to ∼0.1 Ma their records in Fig. 3 and support the idea that the North Pacific warmed over the Plio-Pleisto transition while the North Atlantic cooled. At the multi-0.1 Ma timescale, such a conclusion should be robust without delving into age models too far.

In general, the paper needs some heavy editing on the writing side for clarity and grammar. For instance, conjunctions, such as "aren't", are used at points.

---

## Referee Comment (RC2) · Anonymous Referee #2 · 14 May 2019

Sánchez-Montes et al. present a new comprehensive set of Plio-Pleistocene records from IODP site U1417 in the Gulf of Alaska encompassing SST, IRD, input of terrigenous organic matter and pollen counts. The authors infer dynamics of the Cordilleran ice sheet over 4-1.7 Ma and discuss conceptual models for potential climatic controls. It is an exciting dataset and a valuable contribution to the debate on regional versus global climatic triggers for glaciation in the Northeast-Pacific realm during the Northern Hemisphere Glaciation. The study also adds new information to the plioVar database. The application of biomarkers, pollen and IRD is robust and state of the art. Nevertheless, the manuscript needs some revision regarding the clarity and logic of several parts in the discussion.

[Figure]

The interpretations of the TAR-index partly need a more detailed discussion to clarify the interactions of different factors controlling the TAR (i.e. vegetation cover, petrogenic contributions and aquatic production) and the link to glaciation. At the present stage, particularly section 4.3 on the iNHG and the early Pleistocene is inconclusive with respect to variations in the sources of organic matter and the inferences on glaciation dynamics in the region. Also, the chronology of processes described in section 4.3 is a bit convoluted. In order to clarify and strengthen the interpretations of the TAR-index, the CPI has to be better represented in the manuscript. At the moment it is mentioned a few times in the text but the record is not shown in any figure. I recommend to plot the CPI along with the TAR in figure 2.

In section 4.4 the discussion about the climatic controls on glaciation is very hard to follow and needs to be revisited for clarity. The reader gets lost in the detailed descriptions and comparisons of different gradients during the Plio-Pleistocene and today. I recommend to at least shorten these paragraphs or to delete them. Similarly, the extensive discussion of the PDO analogue in cold and warm periods is confusing and could be shortened. Since the Plio-Pleistocene SST gradients are highly dependent to uncertainties in the absolute SST-estimates associated with the application of UK'37, the gradients need to be discussed in context of those uncertainties. In light of uncertainties on absolute values, section 4.4 would be strengthened by setting the focus on the warming trend that is recorded across the entire North Pacific instead of setting it on the SST gradients.

Moreover, the manuscript should be revisited in terms of language and grammar. There are several spelling and grammar mistakes throughout the manuscript (see detailed comments below). This also applies to the supplementary material.

Detailed comments:

p. 1, line 24: MPWP should be called Mid-Piacenzian-Warm-Period.

p. 5, line 19: "…provides similar SST estimates in northern high latitudes than previous

calibrations." Replace "than" by "to".

p. 5, lines 17-20. I recommend to mention the standard errors of the calibrations.

p.5, line 23: It would be helpful for non-biomarker experts to indicate what the authors wish to reconstruct using the %C37:4.

p.5, line 29: Sentence uses present tense. Turn to simple past.

p. 7, line 1: What is the standard deviation of the statistical mean?

p. 7, line 5: "Gi1 period (3.6-3.4 Ma) warm period…": I suggest to write …"Gi1 warm period (3.6-3.4 Ma)" or something similar along this line.

p. 7 lines 22-24: It is not clear how the high TAR-values relate to limited mountain glaciation as the interpretation of the TAR is missing. The same applies to the %C37:4.

p. 11, lines 1-3: What about petrogenic contributions?

p. 11, line 7: Which interval is meant by: "at first"?

p. 11, line 9: How does the erosion pattern explain the TAR? I don't understand which TAR-variations the authors address.

p. 11, line 12: Do the authors mean an "alternative or additional explanation" to the interpretations in lines 1-3?

p. 11, lines 12-15: which changes in the TAR do the authors mean? Do they refer to the iNHG or the period afterwards? Does the CPI record a change in the source?

p.11, lines 16-17: when exactly is this change in the CPI recorded? How is the switch in the source "away from the more mature coal bedrock" connected to the Surveyor Channel? Does it mark a switch to the channel or a switch away from the channel?

p. 11, lines 20-21: I recommend to add a standard deviation to the average values.

p. 13, line 9: decree or degree?

p. 13, line 10: "aren't" should be "are not".

p.13, lines 11-13: the reference to the figures seems to be mixed up here. C is indicated as summer in the text while in figure 1 panel C is references as winter.

p.14, lines 21-22: how do the vegetation reconstructions from this study fit the results deduced from the El'Gygytgyn pollen record?

p.14, line 32, "the data is the first climatic data": replace "is" by "are".

Figure 1: The sites can be larger and I also suggest to add the study site U1417 to panel A.

Figure 2 and 3: I recommend to increase the size of these figures. They show a lot of data and the small size makes them look quite busy. It is sometimes hard to read the small annotations. I suggest to increase the font size and also the lengths of the x-axes. Some graphs overlap each other as the y-axes are very closely spaced. The distances between the y-axes should be increased a bit. The x-axes would be easier to read if minor ticks were shown. In Figure 3 the line thickness of the x-axis should be increased and I suggest to add data points to the single graphs, as done in Figure 2.

---

## Editor Comment (EC1) · Alberto Reyes (Editor) · 4 Jun 2019

Dear Dr. Sánchez-Montes and co-authors,

The discussion period for your manuscript is over, and two reviewers have posted comments. Both reviewers were positive about the multiproxy dataset from Site U1417 and the potential for insight into late Pliocene glaciation in the Gulf of Alaska region, and I agree that the data and manuscript will be interesting to the readership of Climate of the Past.

However, there are several broad points of concern. In particular, both reviewers iden-

tify section 4.4 as problematic with respect to clarity and whether the data support the conclusions being made. I also agree that the manuscript would benefit from additional support for the biomarker-based inferences on glaciation and some discussion of the published literature on terrestrial records of late Pliocene CIS glaciation. I think it should be possible to address these concerns with moderate-to-major revision of the manuscript.

Please respond to the reviewer comments in the online Discussion forum. If you intend to make any changes to your manuscript in response to these comments, please clearly indicate the nature of these changes in your response. Once I have reviewed your response, I fully anticipate inviting you to submit a revised manuscript for further consideration.

Sincerely, Alberto Reyes

---

## Author Comment (AC2) · 22 Jun 2019

We would like to thank Anonymous Referee 2 for the constructive comments provided to help us improve our manuscript. Please find below our responses to these comments and the manuscript changes.

1-Comment from Referee

Sánchez-Montes et al. present a new comprehensive set of Plio-Pleistocene records from IODP site U1417 in the Gulf of Alaska encompassing SST, IRD, input of terrigenous organic matter and pollen counts. The authors infer dynamics of the Cordilleran

ice sheet over 4-1.7 Ma and discuss conceptual models for potential climatic controls. It is an exciting dataset and a valuable contribution to the debate on regional versus global climatic triggers for glaciation in the Northeast-Pacific realm during the Northern Hemisphere Glaciation. The study also adds new information to the plioVar database. The application of biomarkers, pollen and IRD is robust and state of the art. Nevertheless, the manuscript needs some revision regarding the clarity and logic of several parts in the discussion. The interpretations of the TAR-index partly need a more detailed discussion to clarify the interactions of different factors controlling the TAR (i.e. vegetation cover, petrogenic contributions and aquatic production) and the link to glaciation. At the present stage, particularly section 4.3 on the iNHG and the early Pleistocene is inconclusive with respect to variations in the sources of organic matter and the inferences on glaciation dynamics in the region. Also, the chronology of processes described in section 4.3 is a bit convoluted. In order to clarify and strengthen the interpretations of the TAR-index, the CPI has to be better represented in the manuscript. At the moment it is mentioned a few times in the text but the record is not shown in any figure. I recommend to plot the CPI along with the TAR in figure 2.

2-Author's response

We agree that adding a more detail discussion on the TAR sources and associations with climate would help to deliver a clearer message in our manuscript. We will include details as suggested (see replies on your detailed comments at the end of the document). The authors agree that including the CPI record in Fig. 2 in the manuscript would help in visualising its variations, including arrows where organic matter becomes more/less mature. We will amend Fig. 2 to include the CPI (see Fig. 1 below) and we will present some information on the broad changes in organic matter sources to the GOA (since the data shows a slight increase towards less mature OM as IRD inputs increase). As the reviewer notes (and as we noted in our reply to reviewer 1), there are multiple potential sources to the TAR, including complex onshore petrogenic sources (Yakutat terrain; Childress, 2016), which are not easily disentangled. In the revised

manuscript we can comment that we see evidence for less mature organic matter (increase in CPI) contributing as ice-rafting increases and the TAR decreases, suggesting a potential shift in organic matter source as the glaciation develops (also becomes evident by the range of C and N bulk and isotopes at Site U1417 that show a shift during the early Pleistocene in comparison with the Pliocene and NHG, Table 1 below).

3-Author's changes in manuscript

We will amend Fig. 2 to include the CPI (see Fig. 1 below) and we will present some information on the broad changes in organic matter sources to the GOA (since the data shows a slight increase towards less mature OM as IRD inputs increase).

1-Comment from Referee

In section 4.4 the discussion about the climatic controls on glaciation is very hard to follow and needs to be revisited for clarity. The reader gets lost in the detailed descriptions and comparisons of different gradients during the Plio-Pleistocene and today. I recommend to at least shorten these paragraphs or to delete them. Similarly, the extensive discussion of the PDO analogue in cold and warm periods is confusing and could be shortened. Since the Plio-Pleistocene SST gradients are highly dependent to uncertainties in the absolute SST-estimates associated with the application of UK'37, the gradients need to be discussed in context of those uncertainties. In light of uncertainties on absolute values, section 4.4 would be strengthened by setting the focus on the warming trend that is recorded across the entire North Pacific instead of setting it on the SST gradients.

2-Author's response

This was also a concern of Reviewer 1. The authors agree that this section needs to be shortened to avoid complexity.

3-Author's changes in manuscript

We will implement your comments and edit this section accordingly. In addition, we

have smoothed down to 100 kyr the data sets in Figure 3 in the manuscript following Reviewer 1's advice (see Figure 2 attached).

The text in section 4.4 will be edited (shortened and clarified) to read as follows: "The overall cooling trend during the Neogene, briefly interrupted by the MPWP and intense cooling events such as the M2, is believed to be a dominant pattern in the global climate. This notion is largely based on the global increase in ice volume (e.g. LR04 Benthic ïĄď18O Stack (Lisiecki and Raymo, 2005) and from studies in the North Atlantic SST (i.e. ODP Site 982, Lawrence et al., 2009). In contrast, the contribution of the North Pacific into our understanding of the global climate evolution from the Pliocene to the Pleistocene is limited. Our study at Site U1417 adds valuable regional climate information during the evolution of the Cordilleran Ice Sheet. Unlike the LR04 stack, average Pliocene SST values (4.0 to 2.8 Ma) at Site U1417 are 1 °C colder than the average early Pleistocene values (2.7 to 1.7 Ma) (the Pliocene-Pleistocene SST difference of 1°C has an standard deviation of 0.5°C). In the wider North Pacific, a warming trend from the late Pliocene to early Pleistocene has also been observed at ODP Site 882 in the subarctic Pacific (Martínez-García et al., 2010), at Site 1010 and potentially at Site 1021 (mid-latitude east Pacific) (Fig. 3). Beyond the North Pacific, warmer SST during the early Pleistocene compared to the Pliocene have also been recorded i.e. DSDP Site 593 in the Tasman Sea (McClymont et al., 2016) and Site 1090 (Martínez-García et al., 2010) in the South Atlantic. In contrast, long-term cooling trends mark the early Pleistocene for the mid-latitude west Pacific (Site 1208) and tropical east Pacific (Site 846), more consistent with the development of a cooler and/or more glaciated climate (Fig. 3).

The North Pacific warming occurs despite an atmospheric CO2 drop from 280-450 ppmv to 250-300 ppmv (similar to pre-industrial levels) from 3.2 to 2.8 Ma (Pagani et al., 2010; Seki et al., 2010) and an associated reduction in global radiative forcing (Foster et al., 2017). The early Pleistocene warming signal in the GOA (and the north Pacific more generally) thus implies an important role for local or regional processes.

We have discussed above the potential role played by ocean stratification in the North Pacific, and a possible link to the evolving Cordilleran Ice Sheet in the GOA through evaporation/precipitation feedbacks. The synchrony of these changes with observed tectonic uplift (e.g. Enkelmann et al. 2015) makes it difficult to disentangle the potential climatic and tectonic mechanisms behind ice sheet expansion.

To understand the evolution of the ocean currents governing the North Pacific at the present core sites (Fig. 1) and to find possible explanations of the observed SST distributions during the Pliocene and Pleistocene climate evolution, the modern climate system is used here as an analogue. Modern monthly mean SSTs at ODP 882 SSTs are colder than Sites U1417 and 1021 all year around. During the late Pliocene and early Pleistocene, ODP 882 SSTs are 3-4 °C warmer than in the east (Fig. 3f and g). Modern seasonal climate analogues cannot be used to explain to Pliocene and Pleistocene subarctic SST distribution. However, on longer timescales, the strength of the AL is currently linked to the wider Pacific Ocean circulation by the Pacific Decadal Oscillation (PDO) over periods of 20-30 years (Furtado et al., 2011). The Pliocene-Pleistocene North Pacific SST gradients show similarities with the negative phase of the PDO (-PDO), which is characterized by positive SST anomalies in the central North Pacific surrounded by negative SST anomalies along the North American coast and in the east equatorial Pacific. The -PDO associated route of winds might have increased the precipitation in the Gulf of Alaska and represent a key factor for the fast building of ice in the Alaskan mountains."

1-Comment from Referee

Moreover, the manuscript should be revisited in terms of language and grammar. There are several spelling and grammar mistakes throughout the manuscript (see detailed comments below). This also applies to the supplementary material.

2-Author's response

Thank you.

3-Author's changes in manuscript

We will review the manuscript for typos and amend the ones that you highlight below.

1-Comment from Referee

p. 1, line 24: MPWP should be called Mid-Piacenzian-Warm-Period.

2-Author's response and changes in manuscript

We will change this.

1-Comment from Referee

p. 5, line 19: ". . .provides similar SST estimates in northern high latitudes than previous calibrations." Replace "than" by "to".

2-Author's response and changes in manuscript

We will change this.

1-Comment from Referee

p. 5, lines 17-20. I recommend to mention the standard errors of the calibrations.

2-Author's response and changes in manuscript

We will mention this in p.5 line 18: "which accuracy is constrained by an standard error of $\pm1.5\,^{\circ}$C" and in p.5 line 23: "The standard error of Prahl et al. (1988) (Eq. (4)) is $\pm1.0\,^{\circ}$C".

1-Comment from Referee

p.5, line 23: It would be helpful for non-biomarker experts to indicate what the authors wish to reconstruct using the %C37:4.

2-Author's response and changes in manuscript

We will explain this in p.5 line 24 "The %C37:4 represents fresher and cooler surface

water characteristics (Bendle et al., 2005). In the Nordic Seas this has been linked to subpolar and polar water masses (Bendle et al., 2005), whereas elsewhere in the North Atlantic it has been linked to freshwater inputs (e.g. during Heinrich events, by Martrat et al., 2007). In the subarctic Pacific, the %C37:4 proxy has been less well studied, but high %C37:4 is also proposed to reflect cooler and fresher water masses (Harada et al., 2006)."

1-Comment from Referee

p.5, line 29: Sentence uses present tense. Turn to simple past.

2-Author's response and changes in manuscript

We will change this.

1-Comment from Referee

p. 7, line 1: What is the standard deviation of the statistical mean?

2-Author's response and changes in manuscript

We will include this value in the manuscript.

1-Comment from Referee

p. 7, line 5: "Gi1 period (3.6-3.4 Ma) warm period. . .": I suggest to write . . ."Gi1 warm period (3.6-3.4 Ma)" or something similar along this line.

2-Author's response and changes in manuscript

We will change this.

1-Comment from Referee

p. 7 lines 22-24: It is not clear how the high TAR-values relate to limited mountain glaciation as the interpretation of the TAR is missing. The same applies to the %C37:4.

2-Author's response and changes in manuscript

We will explain this including the following sentences in p.5 line 5 "Terrigenous and aquatic organic matter sources increase during the early Pleistocene in comparison with the late Pliocene. High TAR values can be indicative of relative increases in terrigenous organic matter transported to the ocean and/or to relative decreases in aquatic microorganism production. The opposite could explain low TAR values. To disentangle the old organic matter contamination from the fresh signal, we include the CPI index (Bray and Evans, 1961). High CPI values indicate a fresher or relatively newly produced organic matter transported to the ocean. CPI close to 1 indicate mature or old organic matter sources, such as coal or oil deposits, eroded to the ocean. This distinction may be important in the GOA, where the onshore bedrock includes units with high contents of terrigenous organic matter (e.g. the Yakutat Terrain, Childress, 2016; Walinsky et al., 2009)."

1-Comment from Referee

p. 11, lines 1-3: What about petrogenic contributions?

2-Author's response and changes in manuscript

In response to comments from reviewer 1 about organic matter source, and in response to the previous comment, we aim to have given additional detail on the possibility of petrogenic contributions. We will also include this sentence "The CPI values discard mature sources of organic matter to the GOA at this time interval suggesting a contemporary aquatic organic matter contribution."

1-Comment from Referee

p. 11, line 7: Which interval is meant by: "at first"?

2-Author's response and changes in manuscript

We will change this to read "This could indicate that the first IRD in icebergs delivered to the GOA during the late Pliocene and early Pleistocene originated from smaller marine terminating valley glaciers which removed sediment and weathered rock from

the landscape rather than eroding bedrock and allowed IRD generation."

1-Comment from Referee

p. 11, line 9: How does the erosion pattern explain the TAR? I don't understand which TAR-variations the authors address.

2-Author's response and changes in manuscript

We have deleted this sentence as the idea is better expressed in the previous sentence (see previous comment).

1-Comment from Referee

p. 11, line 12: Do the authors mean an "alternative or additional explanation" to the interpretations in lines 1-3?

2-Author's response and changes in manuscript

This is now changed to say "Additional" only.

1-Comment from Referee

p. 11, lines 12-15: which changes in the TAR do the authors mean? Do they refer to the iNHG or the period afterwards? Does the CPI record a change in the source?

2-Author's response and changes in manuscript

We have made this clearer by adding a time reference in the sentence: "An additional explanation for the changing TAR during the early Pleistocene is that tectonic uplift of the Chugach/St Elias area from 2.7 Ma (Enkelmann et al., 2015) led to orogenic precipitation and a change in erosional pathways (Enkelmann et al., 2015)." Then this sentence link to the CPI values mentioned in the next sentence over the same time period.

1-Comment from Referee

p.11, lines 16-17: when exactly is this change in the CPI recorded? How is the switch in the source "away from the more mature coal bedrock" connected to the Surveyor Channel? Does it mark a switch to the channel or a switch away from the channel?

2-Author's response and changes in manuscript

Site U1417, which is located in the surveyor Channel, contains organic matter with a different provenance: terrigenous from different sources (vegetation and different land sediments or aquatic (i.e. phytoplankton). We will make this change more concrete by adding in line 15: "An increase in CPI variability to concentrations up to 2 and 3 during the early Pleistocene (starting from 2.7 Ma) supports the change of source of organic matter away from the more mature coal bedrock into more immature terrestrial organic matter (plant waxes). However, this comes at a time of increasing IRD, which adds a new source of terrigenous sediment to Site 1417. The shift in CPI values at 2.7 Ma agrees with the shift towards the erosion of sediments sourced from metamorphic and plutonic sources, described in Enkelmann et al. (2015) delivered to Site U1417."

1-Comment from Referee

p. 11, lines 20-21: I recommend to add a standard deviation to the average values.

2-Author's response and changes in manuscript

We will change this.

1-Comment from Referee

p. 13, line 9: decree or degree?

2-Author's response and changes in manuscript

We have now erased this sentence as part of the shortening of section 4.4 requested.

1-Comment from Referee

p. 13, line 10: "aren't" should be "are not".

2-Author's response and changes in manuscript

We have now erased this sentence as part of the shortening of section 4.4 requested.

1-Comment from Referee

p.13, lines 11-13: the reference to the figures seems to be mixed up here. C is indicated as summer in the text while in figure 1 panel C is references as winter.

2-Author's response and changes in manuscript

We have now erased this sentence as part of the shortening of section 4.4 requested.

1-Comment from Referee

p.14, lines 21-22: how do the vegetation reconstructions from this study fit the results deduced from the El'Gygytgyn pollen record?

2-Author's response and changes in manuscript

We have deleted this reference during the shortening of section 4.4 requested.

1-Comment from Referee

p.14, line 32, "the data is the first climatic data": replace "is" by "are".

2-Author's response and changes in manuscript

We have now erased this sentence as part of the shortening of section 4.4 requested.

1-Comment from Referee

Figure 1: The sites can be larger and I also suggest to add the study site U1417 to panel A.

2-Author's response and changes in manuscript

We will change this.

1-Comment from Referee

Figure 2 and 3: I recommend to increase the size of these figures. They show a lot of data and the small size makes them look quite busy. It is sometimes hard to read the small annotations. I suggest to increase the font size and also the lengths of the x-axes. Some graphs overlap each other as the y-axes are very closely spaced. The distances between the y-axes should be increased a bit. The x-axes would be easier to read if minor ticks were shown. In Figure 3 the line thickness of the x-axis should be increased and I suggest to add data points to the single graphs, as done in Figure 2.

2-Author's response and changes in manuscript

We will change this.

————————————————

References cited in our reply:

Addison, J., A., Finney, B. P., Dean, W. E., Davies, M. H., Mix, A. C., Stoner, J. S. and Jaeger, J. M., 2012, Productivity and sedimentary d15N variability for the last 17,000 years along the northern Gulf of Alaska continental slope, Paleoceanography, Vol 27, PA 1206.

Childress, L. B., 2016, The Active Margin Carbon Cycle: Influences of Climate and Tectonics in Variable Spatial and Temporal Records, PhD thesis Northwestern University, Evanson, Illinois.

————————————————————————

[Figure]

**Fig. 1.** Figure 1 TAR, CPI, SR and IRD at Site U1417. Missing data points are either a result of samples analysed for SSTs at the early stages of the project which were not subsequently analysed for n-alkanes.

[Figure]

**Fig. 2.** Figure 2: ∼100 Kyr smoothed North Pacific sites (adapted from Fig. 3 in original manuscript).

| | EW0408–85JC | Site U1417 | | |
|---|---|---|---|---|
| | | Pliocene | NHG | Early Pleistocene |
| N/C | 0.035 to 0.12 | 0.06 to 0.26 | 0.05 to 0.23 | 0.04 to 0.13 |
| $\delta^{13}C$ (‰) | -26.5 to -22 | -26.0 to -21.8 | -25.9 to -23 | -25.35 to -23.9 |

**Fig. 3.** Table 1: N/C vs $\delta 13C$ (‰) at Site U1417 vs range of data at EW0408–85JC (Addison et al., 2012). Data from the Pliocene (4 to 3 Ma), NHG (2.9 to 2.4 Ma) and the early Pleistocene (2.3-1.7 Ma).

---

## Author Comment (AC3) · 22 Jun 2019

[revised manuscript text omitted]

---

## Author Response (AR1)

We would like to thank Anders Carlson for the comments provided to encourage discussion of how our manuscript could be improved. Please find below our responses to these comments and the changes in the manuscript.

1-Comment from the referee
Sánchez-Montes et al. present new and exciting SST and IRD data from IODP Site U1417 in the Gulf of Alaska spanning 4 to 1.7 Ma. This occurs during the Pliocene to Pleistocene transition when the world shifted into the modern ice age period. By measuring IRD and SST records in the same core, the authors can provide a one-to- one comparison between local temperature and the behavior of tidewater glaciers. While this data is exciting, the paper needs some substantial revisions and consideration of other data not referenced. I'll leave my comments at a more general level at present to guide revision prior to more specific comments.
First, don't call it the mid-Pliocene warm period. The MPWP is actually the mid-Piacenzian warm period. The Piacenzian used to be in the mid Pliocene but as of 2009, it is now the late Pliocene. See Gibbards et al.'s revision of the Cenozoic timescale.

2-Author's response
Thank you for spotting this, we agree this should be changed.

3-Author's changes in the revised manuscript
The MPWP is amended in a revised manuscript to be the acronym of the Mid-Piacenzian Warm Period (see page 1, line 24).

1-Comment from the referee
Second, the authors should skip this comparison with modern SST in their discussion and rather focus on comparing with the alkenone record of Praetorius et al. (2015 Nature) that is from 17 ka-0 ka at what is now Site U1419 (core is EW0408-85JC taken on the site survey cruise for the IODP leg). The core top temp is the same as at U1417, supporting the comparison. And more importantly, the glacial interglacial absolute and relative change at U1419 from 17 to 0 ka is the same as the Pliocene and early Pleistocene range and absolute temps at U1417. I think this rules out a major change in CO2 average composition as a driver of a Pliocene to Pleistocene to last deglaciation SST pattern.

2-Author's response
We could acknowledge the similarity in the range of glacial-interglacial SST changes between the two sites in the manuscript, but with the strong caveat that we would not be comparing like with like. We would like to highlight that U1417 and U1419 are 400 km apart and have around 4 km difference in water depth. As we show on Figure 1 in our original manuscript, presently these sites are under the influence of different oceanic currents: U1417 is influenced by the Alaskan Current and U1419 is influenced by the Alaskan Coastal Current. If we choose to compare U1417 during the 4 to 1.7 Ma time period with U1419 covering the 17 Ka to 0 Ka interval, we would be comparing the behaviour change over time of two different currents (and we do not have LGM- Holocene data from U1417 to make a direct comparison to

U1419). We do not see the benefit of this comparison as an alternative to our current comparison between the modern SSTs at our core site with those of the Pliocene and early Pleistocene. In our manuscript, we prefer to compare Site U1417 with modern SST at the same location (Site U1417), to give information about the degree of change of that particular current under different past climates.

3-Author's changes in the revised manuscript
Reflecting on the reviewer's comments, a sentence in the revised manuscript has been amended to highlight the modern location of Site U1417:

"The location of Site U1417 rests under the modern influence of the AC (Fig. 1)." (Page 3, line 17).

Two additional sentences clarify the benefit of the Pliocene and Pleistocene-present comparison in the revised manuscript:

"We compare our palaeo-SST with the modern SST (here "modern" refers to the averaged decadal statistical mean SST of 6.5 °C (standard deviation of 3.4 °C) during the 1955 to 2012 time period, NOAA WOD13; Boyer *et al*., 2013) at the location of Site U1417 to observe changes in the behaviour of the Alaskan Current." (Page 7, lines 17-20)

"To understand the evolution of the ocean currents governing the North Pacific at the present core sites (Fig. 1) and to find possible explanations of the observed SST distributions during the Pliocene and Pleistocene climate evolution, the modern climate system is used here as an analogue." (Page 13, lines 24-26).

1-Comment from the referee
Third, quit saying this is the Cordilleran ice sheet. IRD only means you have a marine terminating ice margin. There is no Cordilleran ice sheet today and the Gulf of Alaska has a lot of marine terminating ice margins and icebergs floating around. For instance, both Bering and Malaspina glaciers could quickly become marine terminating if a big storm came through and blew away their morainal banks that happen to right now be above sea level. Both glaciers have beds below sea level except for that little bank. The evidence for an ice sheet comes from the dated proglacial gravels to the northeast in Hidy et al. (QSR 2013) that date the first Cordilleran ice sheet to about 2.6 Ma. This paper should be discussed. Likewise, the authors should use the proximal mag sus. record from ODP Site 887 rather than the far removed to western Pacific records of 882 that Haug et al. produced. I would also include comparison of the U1419 mag sun record to 887 to support the authors suggestions/conclusions. The mag sus record could also help in improving the IRD resolution/interpretation. Surprised it isn't included. In summary, the authors should just refer to tidewater glaciation of the mountains, leaving out the word ice sheet or Cordilleran ice sheet. As far as the record they have in U1419, the conclusion is that some icebergs survived to U1419 once at 2.7 Ma and then again after 2.6 Ma. This is important findings but by no means says anything about an ice sheet or its size. The authors could compare IRD abundance to the IRD record from Addison et al.

(2012 Paleoc) on 85JC. Now, 85JC is much closer to the ice margin and coast but could provide some kind of context.

2-Author's response

As we noted in the previous comment, Site U1417 is located at present ~450 km away from the coast. At present, tidewater glaciers have retreated far inland from their advance during the Little Ice Age (Molnia, 2007; 2008). Although icebergs calve into fjords and bays today, they do not survive to reach the Gulf of Alaska. We have found literature on the heavy influence of glacier runoff on the characteristics of the ACC that flows along the NW Alaskan coast (Weingartner et al., 2005; Royer and Grosch, 2006), which today acts as a barrier to icebergs reaching central GOA. Thus, the enhanced iceberg delivery into the GOA during the early Pleistocene requires a more extensive and/or productive calving margin in ice from the Cordillera, than is observed today.

In our multi-proxy data set from Site U1417 we observe notable changes, in addition to peaks in IRD between 3 to 2.5 Ma that support the establishment of the Cordilleran Ice Sheet (CIS). These include a step-wise increase in sedimentation rates, increase meltwater discharge and increase in the delivery of terrestrial leaf-wax lipids. We assert that the observations recorded at U1417 from 3-2.5 Ma indicate widespread glaciation along the Alaskan margin resulting from an expanding CIS with marine-ending outlet glaciers, and not an advance of a singular mountain glacier to tidewater. A previous study (St John and Krissek, 1999) also identified IRD increasing in Site 887, which is located 200 km southwest of U1417. In Figure 2, IRD from both Site 887 and U1417 are plotted. Site U1417 records a higher abundance of IRD than Site 887, which we infer could be due to its proximity to the ice in coastal Alaska (Fig. 1). Mindful of the differences in the age models of the two sites, the IRD peaks show similar increases and decreases during the 4 to 1.7 Ma interval, suggesting a wider distribution of enhanced iceberg delivery to the GOA than might be expected from a single outlet glacier.

Our study advances the understanding of ice rafting in the North Pacific (as discussed in St. John and Krissek, 1999) by considering SSTs as a significant factor in the survivability of icebergs transiting the Pacific Ocean (page 11, lines 24-28 in the revised manuscript). For example, a reduction in SSTs occurs in association with an IRD peak at 2.9 Ma, and several high frequency peaks in IRD from 2.7 to 2.4 Ma are associated with higher and more variable SSTs. The reviewer notes the work of Hidy et al. (2013) and their chronology for the Klondike gravel, which marks the 'earliest and most extensive Cordilleran ice sheet' on its eastern margins. Hidy et al. (2013) use independent cosmogenic nuclide dating to identify the maximum advance of the CIS at 2.64 Ma (+0.20/-0.18 Ma). A large ice advance on land in the eastern Cordillera, at the same time as enhanced IRD delivery to two sites in the North Pacific (U1417, ODP 887) suggest that widespread glaciation had developed in the interval 2.7-2.4 Ma.

Icebergs were calved from the CIS marine-terminating outlet glaciers, which cut troughs across the continental shelf (e.g. shown for the Last Glacial Maximum (LGM) in Fig 1 of Gulick et al., 2015). As noted during the original IODP expedition, the dominance of low-grade metamorphic lithologies suggest that the Chugach metamorphic complex (Jaeger et al., 2014) was a primary source of IRD for Lithological Unit I (Jaeger et al., 2014). Due to the spatial extent of this formation and without further detailed investigation of clast lithologies (and their geochemistry) it is not possible for us to test whether the IRD being deposited at Site U1417 is from one tidewater glacier system or not, although it is not clear why one glacier would advance and generate sufficient IRD in isolation, especially in the context of the

eastward advance onshore detailed by Hidy et al. (2013) discussed above. It is beyond the scope of this study to test this question. However, the TAR, CPI and IRD Rc data we also generated for Site U1417 sediments suggest a mix of sediment sources being eroded, transported and deposited to Site U1417. This further supports an influence of widespread glaciation influencing Site U1417 during the Cordilleran Ice Sheet expansion. Connecting this comment with a comment from Reviewer 2, we propose that including the CPI (carbon preference index) might help in visualising the diversity of material (indicated by the maturity of the sediments) reaching Site U1417 and thus, the diverse provenance of material eroded and transported to Site U1417. The distance of U1417 from the present coastline and the diversity of material arriving to Site U1417 from the terrain strongly suggests a mixed source of organic matter to the Site.

We have cited Gulick et al., 2015 in order to link Site U1417 to the other sites (including U1419) drilled along a transect by IODP Expedition 341. This transect extends from the deep sea to the Alaskan Margin. It is important to note that the high sedimentation rates in this temperate glacimarine setting preclude the direct comparison of the oNHG described in our paper with any other sites drilled closer to the continent (including U1419 and 85JC mentioned by the referee, which do not extend beyond the most recent glacial stage). The response of CIS deglaciation after the LGM has been documented at 85JC on the continental rise by sudden reductions in sedimentation rate, IRD delivery, and bulk density (Davies et al., 2011) but the record does not extend to the onset of glaciation during the last marine isotope stage. In response to the reviewer's concerns, we can include additional comparison of our data for the Pliocene/early Pleistocene at Site U1417 with the late Pleistocene data of Sites U1419 / 85JC, but with the caveat that these are quite different sedimentary environments and timescales (as evident by the different range of sedimentary N/C vs δ13C (‰ ) signature at 85JC and U1417).

3-Author's changes in the revised manuscript
We have replaced the Haug et al., 2005 citation with the Hidy et al., 2013 reference:

"The later onset (oNHG) or intensification (iNHG) of the Northern Hemisphere Glaciation is marked by the expansion of the Laurentide, Greenland and Scandinavian ice sheets around 2.5 Ma indicated by ice rafted debris (IRD) records from the North Atlantic Ocean (i.e. Shackleton *et al*., 1984) and the advance of the Cordilleran Ice Sheet at 2.7 Ma inferred from a terrestrial record (Hidy *et al*., 2013)." (Page 2, line 4)

We have included Hidy et al. (2013)'s reference in our revised manuscript to support the attribution of our IRD advance to the Cordilleran Ice Sheet expansion:

"The timing of the increase in IRD at Site U1417 coincides with the increase in IRD at Site 887 (St John and Krissek, 1999) and the maximum extent of the CIS as recorded onshore in the eastern Cordillera by the extensive Klondike gravels at 2.64 Ma (+0.20/-0.18 Ma) (Hidy et al., 2013)." (page 11, line 25).

We have included Hidy et al, 2013's Lower Klondike Valley and ODP 887 locations in Figure 1b and c.

The age of the CIS glaciation in Yukon interior (Hidy et al., 2013) (purple vertical line) and the IRD MAR record at ODP 887 (Fig. 2d) are now included in Fig. 2.

We have clarified the overall increasing terrestrial organic matter trend through the Pliocene and Pleistocene

"Terrigenous and aquatic organic matter sources increase during the early Pleistocene in comparison with the late Pliocene." (page 5, line 9).

and linked it to the CIS glaciation

"The most reasonable explanation is that the land was becoming increasingly ice covered, so that the erosion of vegetation and terrigenous organic matter eroded and transported to the ocean increased" (page 11, line 4-5).

We have included the possible provenance of TAR:

"High TAR values can be indicative of relative increases in terrigenous organic matter transported to the ocean and/or to relative decreases in aquatic microorganism production. The opposite could explain low TAR values. To disentangle the old organic matter contamination from the fresh signal, we include the CPI index (Bray and Evans, 1961). High CPI values indicate a fresher or relatively newly produced organic matter transported to the ocean. CPI close to 1 indicate mature or old organic matter sources, such as coal or oil deposits, eroded to the ocean. This distinction may be important in the GOA, where the onshore bedrock includes units with high contents of terrigenous organic matter (e.g. the Yakutat Terrain, Childress, 2016; Walinsky *et al*., 2009)." (Page 5, lines 10-16).

We have included the CPI (carbon preference index) in Fig. 2 to help in visualising the diversity of material (indicated by the maturity of the sediments) reaching Site U1417 and thus, the diverse provenance of material eroded and transported to Site U1417.

1-Comment from the referee
Fourth, I would greatly reduce to just cut the discussion of the PDO or analogues to modern SST patterns from the paper. The whole section is very confusing and hard to follow. Likewise, this depends highly on the age models of all the cores and these are not discussed. To make such comparisons/conclusions, common age models and uncertainties need to be applied which I think is beyond the scope of this paper. Rather, the authors should smooth down to âG ̌a ̃0.1 Ma their records in Fig. 3 and support the idea that the North Pacific warmed over the Plio-Pleisto transition while the North Atlantic cooled. At the multi-0.1 Ma timescale, such a conclusion should be robust without delving into age models too far.

2-Author's response
Both reviewers have commented on the complexity of this section and we agree that smoothing the data in Figure 3 of the manuscript and trim the text significantly would result in a clearer description of the North Pacific Plio-Pleistocene climate patterns.

3-Author's changes in the revised manuscript
We have smoothed the data in Figure 3 of the manuscript to highlight more clearly the North Pacific Plio-Pleistocene climate patterns.

The text in section 4.4, has been edited (shortened and clarified) to read as follows:

"The overall cooling trend during the Neogene, briefly interrupted by the MPWP and intense cooling events such as the M2, is believed to be a dominant pattern in the global climate. This notion is largely based on the global increase in ice volume (e.g. LR04 Benthic $\delta^{18}O$ Stack (Lisiecki and Raymo, 2005) and from studies in the North Atlantic SST (i.e. ODP Site 982, Lawrence *et al*., 2009). In contrast, the contribution of the North Pacific into our understanding of the global climate evolution from the Pliocene to the Pleistocene is limited. Our study at Site U1417 adds valuable regional climate information during the evolution of the Cordilleran Ice Sheet. Unlike the LR04 stack, average Pliocene SST values (4.0 to 2.8 Ma) at Site U1417 are 1 °C colder than the average early Pleistocene values (2.7 to 1.7 Ma) (the Pliocene-Pleistocene SST difference of 1°C has an standard deviation of 0.5°C). In the wider North Pacific, a warming trend from the late Pliocene to early Pleistocene has also been observed at ODP Site 882 in the subarctic Pacific (Martínez-García *et al*., 2010), at Site 1010 and potentially at Site 1021 (mid-latitude east Pacific) (Fig. 3). Beyond the North Pacific, warmer SST during the early Pleistocene compared to the Pliocene have also been recorded i.e. DSDP Site 593 in the Tasman Sea (McClymont *et al*., 2016) and Site 1090 (Martínez-García *et al*., 2010) in the South Atlantic. In contrast, long-term cooling trends mark the early Pleistocene for the mid-latitude west Pacific (Site 1208) and tropical east Pacific (Site 846), more consistent with the development of a cooler and/or more glaciated climate (Fig. 3).

The North Pacific warming occurs despite an atmospheric $CO_2$ drop from 280-450 ppmv to 250-300 ppmv (similar to pre-industrial levels) from 3.2 to 2.8 Ma (Pagani *et al*., 2010; Seki *et al*., 2010) and an associated reduction in global radiative forcing (Foster *et al*., 2017). The early Pleistocene warming signal in the GOA (and the north Pacific more generally) thus implies an important role for local or regional processes. We have discussed above the potential role played by ocean stratification in the North Pacific, and a possible link to the evolving Cordilleran Ice Sheet in the GOA through evaporation/precipitation feedbacks. The synchrony of these changes with observed tectonic uplift (e.g. Enkelmann *et al*. 2015) makes it difficult to disentangle the potential climatic and tectonic mechanisms behind ice sheet expansion.

To understand the evolution of the ocean currents governing the North Pacific at the present core sites (Fig. 1) and to find possible explanations of the observed SST distributions during the Pliocene and Pleistocene climate evolution, the modern climate system is used here as an analogue. Modern monthly mean SSTs at ODP 882 SSTs are colder than Sites U1417 and 1021 all year around. During the late Pliocene and early Pleistocene, ODP 882 SSTs are 3-4 °C warmer than in the east (Fig. 3f and g). Modern seasonal climate analogues cannot be used to explain to Pliocene and Pleistocene subarctic SST distribution. However, on longer timescales, the strength of the AL is currently linked to the wider Pacific

Ocean circulation by the Pacific Decadal Oscillation (PDO) over periods of 20-30 years (Furtado *et al.*, 2011). The Pliocene-Pleistocene North Pacific SST gradients show similarities with the negative phase of the PDO (-PDO), which is characterized by positive SST anomalies in the central North Pacific surrounded by negative SST anomalies along the North American coast and in the east equatorial Pacific. The -PDO associated route of winds might have increased the precipitation in the Gulf of Alaska and represent a key factor for the fast building of ice in the Alaskan mountains." (Page 13 and page 14).

1-Comment from the referee
In general, the paper needs some heavy editing on the writing side for clarity and grammar. For instance, conjunctions, such as "aren't", are used at points.

2-Author's response
Thank you.

3-Author's changes in the manuscript
We have carefully reviewed the manuscript to remove typos and improve the clarity in writing.
* * *
References cited in our reply:
Davies, M.H., Mix, A.C., Stoner, J.S., Addison, J.A., Jaeger, J., Finney. B., and Wiest, J. The deglacial transition of the southeastern Alaska Margin: Meltwater input, sea level rise, marine productivity, and sedimentary anoxia. Paleoceanography, Vol. 26, PA2223, 2011.
Jaeger, J.M., Gulick, S.P.S., LeVay, L.J., Asahi, H., Bahlburg, H., Belanger, C.L., Berbel, G.B.B., Childress, L.B., Cowan, E.A., Drab, L., Forwick, M., Fukumura, A., Ge, S., Gupta, S.M., Kioka, A., Konno, S., März, C.E., Matsuzaki, K.M., McClymont, E.L., Mix, A.C., Moy, C.M., Müller, J., Nakamura, A., Ojima, T., Ridgway, K.D., Rodrigues Ribeiro, F., Romero, O.E., Slagle, A.L.,Stoner, J.S., St-Onge, G., Suto, I., Walczak, M.H., and Worthington, L.L., 2014. Site U1417. In Jaeger, J.M., Gulick, S.P.S., LeVay, L.J., and the Expedition 341 Scientists, Proc. IODP, 341: College Station, TX (Integrated Ocean Drilling Program). doi:10.2204/iodp.proc.341.103.2014
Molnia, B.F., 2007. Late nineteenth to early twenty-first century behavior of Alaskan glaciers as indicators of changing regional climate. Global and Planetary Change, v. 56, p. 23-56.
Molnia, B.F., 2008. Glaciers of North America – Glaciers of Alaska: US Geological Survey Professional Paper 1386-K, 525 p.
Prueher, Libby M; Rea, David K: (Table 1) Age, magnetic susceptibility, and mass accumulation rate of volcanic glass and IRD from ODP Site 145-887. PANGAEA, https://doi.org/10.1594/PANGAEA.706309, 2001.
Scharman, R. M., Pavlis, T., Day, E & O'Driscoll, L. (2011). Deformation and structure in the Chugach metamorphic complex, southern Alaska: Crustal architecture of a trans- pressional system from a down plunge section. Geosphere. 10. 10.1130/GES00646.1.

St. John, K.E.K and Krissek, L.A., Regional patterns of Pleistocene ice-rafted debris flux in the North Pacific. Paleoceanography, Vol. 14, 653-662, 1999.

Weingartner, T.J., Danielson, S.L. and Royer, T.C., 2005, Freshwater variability and predictability in the Alaska Coastal Current, Deep-Sea Research Part II-Topical Studies in Oceanography, 52 (1-2): 169-191.

Royer, T. C. and Grosch, C. E., 2006, Ocean Warming and freshening in the northern Gulf of Alaska, Geophysical Research Letters, Vol. 33, L16605, doi:10.1029/2006GL026767, 2006.

We would like to thank Anonymous Referee 2 for the constructive comments provided to help us improve our manuscript. Please find below our responses to these comments and the manuscript changes.

1-Comment from Referee

Sánchez-Montes et al. present a new comprehensive set of Plio-Pleistocene records from IODP site U1417 in the Gulf of Alaska encompassing SST, IRD, input of terrigenous organic matter and pollen counts. The authors infer dynamics of the Cordilleran ice sheet over 4-1.7 Ma and discuss conceptual models for potential climatic controls. It is an exciting dataset and a valuable contribution to the debate on regional versus global climatic triggers for glaciation in the Northeast-Pacific realm during the Northern Hemisphere Glaciation. The study also adds new information to the plioVar database. The application of biomarkers, pollen and IRD is robust and state of the art. Nevertheless, the manuscript needs some revision regarding the clarity and logic of several parts in the discussion. The interpretations of the TAR-index partly need a more detailed discussion to clarify the interactions of different factors controlling the TAR (i.e. vegetation cover, petrogenic contributions and aquatic production) and the link to glaciation. At the present stage, particularly section 4.3 on the iNHG and the early Pleistocene is inconclusive with respect to variations in the sources of organic matter and the inferences on glaciation dynamics in the region. Also, the chronology of processes described in section 4.3 is a bit convoluted. In order to clarify and strengthen the interpretations of the TAR-index, the CPI has to be better represented in the manuscript. At the moment it is mentioned a few times in the text but the record is not shown in any figure. I recommend to plot the CPI along with the TAR in figure 2.

2-Author's response

We agree that adding a more detail discussion on the TAR sources and associations with climate would help to deliver a clearer message in our manuscript. We will include details as suggested (see replies on your detailed comments at the end of the document). The authors agree that including the CPI record in Fig. 2 in the manuscript would help in visualising its variations, including arrows where organic matter becomes more/less mature. We will amend Fig. 2 to include the CPI and we will present some information on the broad changes in organic matter sources to the GOA (since the data shows a slight increase towards less mature OM as IRD inputs increase). As the reviewer notes (and as we noted in our reply to reviewer 1), there are multiple potential sources to the TAR, including complex onshore petrogenic sources (Yakutat terrain; Childress, 2016), which are not easily disentangled. In the revised manuscript we can comment that we see evidence for less mature organic matter (increase in CPI) contributing as ice-rafting increases and the TAR decreases, suggesting a potential shift in organic matter source as the glaciation develops.

3-Author's changes in the revised manuscript

We have amended Fig. 2 to include the CPI and have included some information on the broad changes in organic matter sources to the GOA and ways to explain them in:

"Terrigenous and aquatic organic matter sources increase during the early Pleistocene in comparison with the late Pliocene. High TAR values can be indicative of relative increases in terrigenous organic matter

transported to the ocean and/or to relative decreases in aquatic microorganism production. The opposite could explain low TAR values. To disentangle the old organic matter contamination from the fresh signal, we include the CPI index (Bray and Evans, 1961). High CPI values indicate a fresher or relatively newly produced organic matter transported to the ocean. CPI close to 1 indicate mature or old organic matter sources, such as coal or oil deposits, eroded to the ocean. This distinction may be important in the GOA, where the onshore bedrock includes units with high contents of terrigenous organic matter (e.g. the Yakutat Terrain, Childress, 2016; Walinsky *et al*., 2009)." (Page 5, lines 9-16)

and links to glaciation

"the TAR and CPI values at Site U1417 thus suggest a mix of sources of organic matter during this time dominated by contemporaneous vegetation, although we cannot exclude the possibility of some coal erosion." (absent/limited glaciation) Page 10, lines 29-31.

"From 3 Ma, TAR values decrease to below the average of the entire TAR record, indicating that transport of leaf-wax lipids to Site U1417 increased in comparison with the Pliocene, which may be related to an increase in erosion on land due to the advancing ice-sheet." (Page 10, lines 40-41 and Page 11 line 1).

" The most reasonable explanation is that the land was becoming increasingly ice covered, so that the erosion of vegetation and terrigenous organic matter eroded and transported to the ocean increased."(page 11, lines 4-5)

"The CPI values discard mature sources of organic matter to the GOA at this time interval suggesting a contemporary aquatic organic matter contribution." (Page 12 lines 10-13).

"An additional explanation for the changing TAR during the early Pleistocene is that tectonic uplift of the Chugach/St Elias area from 2.7 Ma (Enkelmann *et al*., 2015) led to orogenic precipitation and a change in erosional pathways (Enkelmann *et al*., 2015). The glaciation could have altered the main source of terrestrial input to the Surveyor Channel, to higher metamorphic and plutonic sources with lower or null TAR values (Childress, 2016). An increase in CPI variability to concentrations up to 2 and 3 during the early Pleistocene (starting from 2.7 Ma) supports the change of source of organic matter away from the more mature coal bedrock into more immature terrestrial organic matter (plant waxes). However, this comes at a time of increasing IRD, which adds a new source of terrigenous sediment to Site 1417. The shift in CPI values at 2.7 Ma agrees with the shift towards the erosion of sediments sourced from metamorphic and plutonic sources, described in Enkelmann et al. (2015) delivered to Site U1417." (Page 12 lines 23-31).

1-Comment from Referee
In section 4.4 the discussion about the climatic controls on glaciation is very hard to follow and needs to be revisited for clarity. The reader gets lost in the detailed descriptions and comparisons of different gradients during the Plio-Pleistocene and today. I recommend to at least shorten these paragraphs or to

delete them. Similarly, the extensive discussion of the PDO analogue in cold and warm periods is confusing and could be shortened. Since the Plio-Pleistocene SST gradients are highly dependent to uncertainties in the absolute SST-estimates associated with the application of UK'37, the gradients need to be discussed in context of those uncertainties. In light of uncertainties on absolute values, section 4.4 would be strengthened by setting the focus on the warming trend that is recorded across the entire North Pacific instead of setting it on the SST gradients.

2-Author's response
This was also a concern of Reviewer 1. The authors agree that this section needs to be shortened to avoid complexity.

3-Author's changes in the revised manuscript
We have implemented your comments and edit this section accordingly. In addition, we have smoothed down to 100 kyr the data sets in Figure 3 in the manuscript following Reviewer 1's advice.

The text in section 4.4 has been edited (shortened and clarified) to read as follows:

"The overall cooling trend during the Neogene, briefly interrupted by the MPWP and intense cooling events such as the M2, is believed to be a dominant pattern in the global climate. This notion is largely based on the global increase in ice volume (e.g. LR04 Benthic $\delta^{18}O$ Stack (Lisiecki and Raymo, 2005) and from studies in the North Atlantic SST (i.e. ODP Site 982, Lawrence *et al.*, 2009). In contrast, the contribution of the North Pacific into our understanding of the global climate evolution from the Pliocene to the Pleistocene is limited. Our study at Site U1417 adds valuable regional climate information during the evolution of the Cordilleran Ice Sheet. Unlike the LR04 stack, average Pliocene SST values (4.0 to 2.8 Ma) at Site U1417 are 1 °C colder than the average early Pleistocene values (2.7 to 1.7 Ma) (the Pliocene-Pleistocene SST difference of 1°C has an standard deviation of 0.5°C). In the wider North Pacific, a warming trend from the late Pliocene to early Pleistocene has also been observed at ODP Site 882 in the subarctic Pacific (Martínez-García *et al.*, 2010), at Site 1010 and potentially at Site 1021 (mid-latitude east Pacific) (Fig. 3). Beyond the North Pacific, warmer SST during the early Pleistocene compared to the Pliocene have also been recorded i.e. DSDP Site 593 in the Tasman Sea (McClymont *et al.*, 2016) and Site 1090 (Martínez-García *et al.*, 2010) in the South Atlantic. In contrast, long-term cooling trends mark the early Pleistocene for the mid-latitude west Pacific (Site 1208) and tropical east Pacific (Site 846), more consistent with the development of a cooler and/or more glaciated climate (Fig. 3).

The North Pacific warming occurs despite an atmospheric $CO_2$ drop from 280-450 ppmv to 250-300 ppmv (similar to pre-industrial levels) from 3.2 to 2.8 Ma (Pagani *et al.*, 2010; Seki *et al.*, 2010) and an associated reduction in global radiative forcing (Foster *et al.*, 2017). The early Pleistocene warming signal in the GOA (and the north Pacific more generally) thus implies an important role for local or regional processes. We have discussed above the potential role played by ocean stratification in the North Pacific, and a possible link to the evolving Cordilleran Ice Sheet in the GOA through evaporation/precipitation feedbacks. The synchrony of these changes with observed tectonic uplift (e.g. Enkelmann *et al.* 2015)

makes it difficult to disentangle the potential climatic and tectonic mechanisms behind ice sheet expansion.

To understand the evolution of the ocean currents governing the North Pacific at the present core sites (Fig. 1) and to find possible explanations of the observed SST distributions during the Pliocene and Pleistocene climate evolution, the modern climate system is used here as an analogue. Modern monthly mean SSTs at ODP 882 SSTs are colder than Sites U1417 and 1021 all year around. During the late Pliocene and early Pleistocene, ODP 882 SSTs are 3-4 °C warmer than in the east (Fig. 3f and g). Modern seasonal climate analogues cannot be used to explain to Pliocene and Pleistocene subarctic SST distribution. However, on longer timescales, the strength of the AL is currently linked to the wider Pacific Ocean circulation by the Pacific Decadal Oscillation (PDO) over periods of 20-30 years (Furtado *et al.*, 2011). The Pliocene-Pleistocene North Pacific SST gradients show similarities with the negative phase of the PDO (-PDO), which is characterized by positive SST anomalies in the central North Pacific surrounded by negative SST anomalies along the North American coast and in the east equatorial Pacific. The -PDO associated route of winds might have increased the precipitation in the Gulf of Alaska and represent a key factor for the fast building of ice in the Alaskan mountains." (Page 13 and page 14).

1-Comment from Referee
Moreover, the manuscript should be revisited in terms of language and grammar. There are several spelling and grammar mistakes throughout the manuscript (see detailed comments below). This also applies to the supplementary material.

2-Author's response
Thank you.

3-Author's changes in the revised manuscript
We have reviewed the manuscript for typos and amend the ones that you highlight below.

1-Comment from Referee
p. 1, line 24: MPWP should be called Mid-Piacenzian-Warm-Period.

2-Author's response and 3- Changes in the revised manuscript
We have changed this in page 1, line 24.

1-Comment from Referee
p. 5, line 19: ". . .provides similar SST estimates in northern high latitudes than previous calibrations." Replace "than" by "to".

2-Author's response and 3-Author's changes in the revised manuscript

We have changed this in page 5 line 33.

1-Comment from Referee
p. 5, lines 17-20. I recommend to mention the standard errors of the calibrations.

2-Author's response and 3-Author's changes in the revised manuscript
We have mentioned this

"which accuracy is constrained by an standard error of $\pm 1.5$ ◦C" (Page 5 lines 31-32)

"The standard error of Prahl et al. (1988) (Eq. (4)) is $\pm 1.0$ ◦C". (Page 6 line 3-4).

1-Comment from Referee
p.5, line 23: It would be helpful for non-biomarker experts to indicate what the authors wish to reconstruct using the %C37:4.

2-Author's response and 3-Author's changes in the revised manuscript
We have explained the C37:4 previous interpretations:

"The $\%C_{37:4}$ represents fresher and cooler surface water characteristics (Bendle *et al*., 2005). In the Nordic Seas this has been linked to subpolar and polar water masses (Bendle *et al.,* 2005), whereas elsewhere in the North Atlantic it has been linked to freshwater inputs (e.g. during Heinrich events, Martrat *et al.*, 2007). In the subarctic Pacific, the $\%C_{37:4}$ proxy has been less well studied (McClymont *et al*., 2008), but high $\%C_{37:4}$ is also proposed to reflect cooler and fresher water masses (Harada *et al*., 2006)." (Page 6 lines 5-9).

And later we interpret it as glacier meltwater sourced at Site U1417:

"$\%C_{37:4}$ increases can be related to colder sea surface conditions, but due to Site U1417's location and climatic context, we suggest that increases in $\%C_{37:4}$ relate to meltwater discharge from the expanding ice-sheet." (Page 10 lines 35-37).

1-Comment from Referee
p.5, line 29: Sentence uses present tense. Turn to simple past.

2-Author's response and 3-Author's changes in the revised manuscript
We have changed this in page 6 line 14.

1-Comment from Referee
p. 7, line 1: What is the standard deviation of the statistical mean?

2-Author's response and 3-Author's changes in the revised manuscript
We have included this value in the revised manuscript in page 7 line 18.

1-Comment from Referee
p. 7, line 5: "Gi1 period (3.6-3.4 Ma) warm period. . .": I suggest to write . . ."Gi1 warm period (3.6-3.4 Ma)" or something similar along this line.

2-Author's response and 3-Author's changes in the revised manuscript
We have changed this in page 7 line 23.

1-Comment from Referee
p. 7 lines 22-24: It is not clear how the high TAR-values relate to limited mountain glaciation as the interpretation of the TAR is missing. The same applies to the %C37:4.

2-Author's response and 3-Author's changes in the revised manuscript
We have explained the interpretation of TAR:

"Terrigenous and aquatic organic matter sources increase during the early Pleistocene in comparison with the late Pliocene. High TAR values can be indicative of relative increases in terrigenous organic matter transported to the ocean and/or to relative decreases in aquatic microorganism production. The opposite could explain low TAR values. To disentangle the old organic matter contamination from the fresh signal, we include the CPI index (Bray and Evans, 1961). High CPI values indicate a fresher or relatively newly produced organic matter transported to the ocean. CPI close to 1 indicate mature or old organic matter sources, such as coal or oil deposits, eroded to the ocean. This distinction may be important in the GOA, where the onshore bedrock includes units with high contents of terrigenous organic matter (e.g. the Yakutat Terrain, Childress, 2016; Walinsky *et al*., 2009)." (Page 5 lines 9-16).

We have amended Fig. 2 to include the CPI.

We have included some information on the broad changes in organic matter sources to the GOA:

[revised manuscript text omitted]

1-Comment from Referee
p. 11, line 7: Which interval is meant by: "at first"?

2-Author's response and 3-Author's changes in the revised manuscript
We have changed this to read:

"This could indicate that the first IRD in icebergs delivered to the GOA during the late Pliocene and early Pleistocene originated from smaller marine terminating valley glaciers which removed sediment and weathered rock from the landscape rather than eroding bedrock and allowed IRD generation." (Page 12, line 17-20).

1-Comment from Referee
p. 11, line 9: How does the erosion pattern explain the TAR? I don't understand which TAR-variations the authors address.

2-Author's response and 3-Author's changes in the revised manuscript
We have deleted this sentence as the idea is better expressed in the previous sentence (see previous comment) and in: "The most reasonable explanation is that the land was becoming increasingly ice covered, so that the erosion of vegetation and terrigenous organic matter eroded and transported to the ocean increased." (Page 11, lines 4-5).

1-Comment from Referee
p. 11, line 12: Do the authors mean an "alternative or additional explanation" to the interpretations in lines 1-3?

2-Author's response and 3-Author's changes in the revised manuscript
This have now changed this to read "Additional" only in page 12 line 23.

1-Comment from Referee
p. 11, lines 12-15: which changes in the TAR do the authors mean? Do they refer to the iNHG or the period afterwards? Does the CPI record a change in the source?

2-Author's response and 3-Author's changes in the revised manuscript
We have made this clearer by adding a time reference in the sentence:

"An additional explanation for the changing TAR during the early Pleistocene is that tectonic uplift of the Chugach/St Elias area from 2.7 Ma (Enkelmann et al., 2015) led to orogenic precipitation and a change in erosional pathways (Enkelmann et al., 2015)." (Page 12 lines 24-25).

Then this sentence link to the CPI values mentioned in the next sentence over the same time period.

1-Comment from Referee
p.11, lines 16-17: when exactly is this change in the CPI recorded? How is the switch in the source "away from the more mature coal bedrock" connected to the Surveyor Channel? Does it mark a switch to the channel or a switch away from the channel?

2-Author's response and 3-Author's changes in the revised manuscript

Site U1417, which is located in the surveyor Channel, contains organic matter with a different provenance: terrigenous from different sources (vegetation and different land sediments or aquatic (i.e. phytoplankton). We have made changes to include:

"An increase in CPI variability to concentrations up to 2 and 3 during the early Pleistocene (starting from 2.7 Ma) supports the change of source of organic matter away from the more mature coal bedrock into more immature terrestrial organic matter (plant waxes). However, this comes at a time of increasing IRD, which adds a new source of terrigenous sediment to Site 1417. The shift in CPI values at 2.7 Ma agrees with the shift towards the erosion of sediments sourced from metamorphic and plutonic sources, described in Enkelmann et al. (2015) delivered to Site U1417." (Page 12, lines 26-31).

1-Comment from Referee
p. 11, lines 20-21: I recommend to add a standard deviation to the average values.

2-Author's response and 3-Author's changes in the revised manuscript
We have included this number:

"(the Pliocene-Pleistocene SST difference of 1°C has an standard deviation of 0.5°C)" (Page 13 lines 7-8).

1-Comment from Referee
p. 13, line 9: decree or degree?

2-Author's response and 3-Author's changes in the revised manuscript
We have now erased this sentence as part of the shortening of section 4.4 requested.

1-Comment from Referee
p. 13, line 10: "aren't" should be "are not".

2-Author's response and 3-Author's changes in the revised manuscript
We have now erased this sentence as part of the shortening of section 4.4 requested.

1-Comment from Referee
p.13, lines 11-13: the reference to the figures seems to be mixed up here. C is indicated as summer in the text while in figure 1 panel C is references as winter.

2-Author's response and 3-Author's changes in the revised manuscript
We have now erased this sentence as part of the shortening of section 4.4 requested.

1-Comment from Referee
p.14, lines 21-22: how do the vegetation reconstructions from this study fit the results deduced from the El'Gygytgyn pollen record?

2-Author's response and 3-Author's changes in the revised manuscript
We have deleted this reference during the shortening of section 4.4 requested.

1-Comment from Referee
p.14, line 32, "the data is the first climatic data": replace "is" by "are".

2-Author's response and 3-Author's changes in the revised manuscript
We have now erased this sentence as part of the shortening of section 4.4 requested.

1-Comment from Referee
Figure 1: The sites can be larger and I also suggest to add the study site U1417 to panel A.

2-Author's response and 3-Author's changes in the revised manuscript
We have changed this in Fig. 1.

1-Comment from Referee
Figure 2 and 3: I recommend to increase the size of these figures. They show a lot of data and the small size makes them look quite busy. It is sometimes hard to read the small annotations. I suggest to increase the font size and also the lengths of the x-axes. Some graphs overlap each other as the y-axes are very closely spaced. The distances between the y-axes should be increased a bit. The x-axes would be easier to read if minor ticks were shown. In Figure 3 the line thickness of the x-axis should be increased and I suggest to add data points to the single graphs, as done in Figure 2.

2-Author's response and 3- Changes in manuscript
We have edited this in Fig. 2 and 3.
* * *
References cited in our reply:
Addison, J., A., Finney, B. P., Dean, W. E., Davies, M. H., Mix, A. C., Stoner, J. S. and Jaeger, J. M., 2012, Productivity and sedimentary d15N variability for the last 17,000 years along the northern Gulf of Alaska continental slope, Paleoceanography, Vol 27, PA 1206.

Childress, L. B., 2016, The Active Margin Carbon Cycle: Influences of Climate and Tectonics in Variable Spatial and Temporal Records, PhD thesis Northwestern University, Evanson, Illinois.
Fig. 1. Figure 1 TAR, CPI, SR and IRD at Site U1417. Missing data points are either a result of samples analysed for SSTs at the early stages of the project which were not subsequently analysed for n-alkanes.
Fig. 2. Figure 2: ~100 Kyr smoothed North Pacific sites (adapted from Fig. 3 in original manuscript).
Fig. 3. Table 1: N/C vs δ13C (‰ ) at Site U1417 vs range of data at EW0408–85JC (Addison et al., 2012). Data from the Pliocene (4 to 3 Ma), NHG (2.9 to 2.4 Ma) and the early Pleistocene (2.3-1.7 Ma).

[revised manuscript text omitted]

---

## Referee Report (RR1)

**2nd review on "Late Pliocene Cordilleran Ice Sheet development with warm Northeast Pacific sea surface temperatures" by Sánchez-Montes et al.**

**General comments**

In my first review I stated to major points of criticism, i.e. the presentation of the TAR and CPI indices and the confusing presentation of section 4.4. The authors addressed both points so that the clarity of the entire discussion greatly improved overall. The new section 4.4 is much clearer than before and the authors' inferences are easy to follow, now. So, I don't have further complaints here. However, I still have some concerns regarding parts of their interpretation of the TAR index which partly arise from having the CPI plotted next to the TAR. At the present stage, the inferences made from the TAR during the oNHG (3-2.8 Ma) are questionable and require further clarification by discussing the different potential factors controlling the TAR in more detail (see below). This is crucial to strengthen the key conclusions that the onset of the Cordilleran Ice Sheet glaciation occurred during this time interval. I am convinced, a more comprehensive discussion around the different factors controlling the TAR will unequivocally strengthen the inferences made from this multi-proxy approach. As already mentioned in my first review, presenting a multi-proxy approach the study is a valuable and important contribution to the ongoing discussion about the glaciation chronology in the Cordillera during the Plio-Pleistocene transition. The paper should be accepted for publication as soon as this last issue regarding the TAR has been addressed.

Interpretation of TAR during the oNHG, 3-2.8 Ma (page 10, line 40-page 11, line 5):

As the authors correctly describe in the methods section, the TAR index estimates the relative abundance of long-chain *n*-alkanes versus the short chain homologues and is commonly used to estimate changes in the relative contributions of plant-wax lipids from higher land plants and aquatic production (page 5, lines 4-7). In the discussion of the onset of the Cordilleran Ice Sheet glaciation (3 to 2.8 Ma) the authors infer an increase in the export of plant-derived organic matter based on a decrease in the TAR-values (page 10, line 40 to page 11, line 5). In fact, this interpretation is contradictory to the general way of interpreting the TAR according to which decreasing values would imply the opposite i.e. an increase in aquatic production and/or a decrease in the contributions from higher land plants. Although I don't doubt the possibility that the export of terrigenous material including the long-chain *n*-alkanes may increase despite a decrease in the TAR, this interpretation needs further justification and cannot be based on the TAR alone. Concentrations or mass accumulation rates of the short-chain and long-chain *n*-alkanes are needed to support this statement since they allow to disentangle the individual developments of the input of short chain and long-chain *n*-alkanes. It may help to compare them to concentrations of other marine biomarkers, e.g. alkenones. Moreover, there are several possibilities to explain the decrease in the TAR-index and not all of them support expanding glaciation in the hinterland. Based on the current presentation of data, the authors don't have a means to rule these options out. Firstly, there could be an increase in aquatic production which does not necessarily mean that the controls on the export of leaf-wax biomarkers change. That means it would be possible that the vegetation cover remains as extensive as before (during the early to mid-Pliocene) which would be in conflict with the authors' inference of advancing ice cover in the region. Secondly, assuming that the

source of the long-chain *n*-alkanes remains the same as during the early to mid-Pliocene, the decreasing TAR may report on a decline in the export of terrigenous organic matter attesting to e.g. a reduction of the vegetation cover. Although this view would be in harmony with advancing ice masses in the hinterland it would be in conflict with the authors' idea of enhanced discharge of leaf-wax lipids from vegetation. Thirdly, the overall low CPI-values (around 1.5) point to a high degree of degradation. So, it is plausible that a large fraction of the long-chain *n*-alkanes may be petrogenic instead of dominantly vegetation-derived. As the authors point out, there are coal-bearing bedrocks in the region and these are characterized by low TAR (up to 2) and low CPI-values (<1) (page 10, lines 25-28 and page 12, lines 25/26). As such, the decrease in the TAR may also indicate intensified erosion of bedrock compared to the early and mid-Pliocene. Under these circumstances, decreasing TAR would be compatible with increased export of terrigenous organic matter. This scenario would be in accordance with the idea of reduced vegetation cover and expanding ice masses in the Cordillera.

Since the inferences for the Ice Sheet Glaciation strongly depend on the way of interpreting the TAR, the different potential controls on the index need a more detailed recognition in the discussion in order to strengthen the authors' key conclusions, namely that the onset of the Cordilleran Ice Sheet Glaciation occurred between 3 and 2.8 Ma.

**Detailed comments**

Page 5, line 9: remove the sentence in line 9. This seems to be out of context here in the method section. The following paragraph encompassing lines 10-16 stands well by itself.

Page 10, line 20: n-alkane => *n*-alkane

Page 12, line 12-13: Actually, the CPI cannot attest to the maturity of the short-chain alkanes as the formula used here does not consider the homologues shorter than $C_{24}$.
The interpretation of the TAR should be justified a little more in detail considering the different interpretation of a similar signal for the period from 3-2.8 Ma. It is not clear why the low TAR values are interpreted as stemming from increased productivity from 2.7 Ma onwards while between 3-2.8 Ma the progressive decrease towards these low values is attributed to an increased export of leaf-wax lipids from land plants. Please, explain.

Page 2, lines 17 and 20: In line 17 it says St. Elias Mountains while in line 20 it is St Elias Mountains. Please, unify the spelling throughout the entire manuscript.

---

## Referee Report (RR2)

Review on Sánchez-Montes et al., 2019

In my last review, my major concern was the interpretation of the TAR-index in section 4.2 (the late Pliocene onset of the Cordilleran Ice Sheet glaciation). The authors inferred increasing terrigenous input from decreasing TAR-values which is contradictory to the definition of the TAR-index where a decrease in the TAR would point to enhanced relative contributions of aquatic production. The inferences on ice sheet dynamics were questionable accordingly. In the new version the authors discuss this section in more detail and by adding mass accumulation rates of short chain and long-chain *n*-alkanes to their data set they provide much clearer and more comprehensive insights into the interactions of different marine and terrestrial processes reflected by their data (i.e. aquatic production, terrigenous organic matter input, vegetation cover and glacial erosion). Their inferences on glaciation dynamics in the Cordillera is well justified and conclusive, now. I only found a few linguistic issues listed below.

I would like to state again that the paper is a valuable contribution to the discussion on NHG adding an important new dataset from the North-Pacific realm. The paper is written in a concise, comprehensible way and the interpretations are well constrained by the data presented. I recommend to accept the paper for publication.

Detailed comments:

Page 10, line 32: remove the comma at the end of the sentence. Moreover, Duk-Rodin => Duk-Rod**k**in.

Page 12, line 4: CIS. I suggest to write "Cordilleran Ice Sheet" as is done in all other instances throughout the paper.

Page 13, line 7: replace "concentrations" by "values"

Page 13, line 15: remove the comma after "(as reflected in the, …."

---

## Author Response (AR2)

We would like to thank Alberto Reyes for his comments provided to help us improve our manuscript. Please find below our responses to these comments and the manuscript changes.

1-Comment from the Editor

Your revised manuscript has now been reviewed by one of the original referees. I agree with the referee that your revisions have resulted in tighter manuscript with improved clarity.

As you can see, the referee provided in-depth and constructive comments regarding interpretation of the TAR data and has recommended major revision. I would like you to consider these comments carefully and submitted another revised manuscript that addresses these concerns. Referee 2 will review your revised submission one more time.

Reviewer 1 (Anders Carlson) asked that you incorporate acknowledgement of the terrestrial record of Cordilleran ice sheet glaciation. It's a good start to include a citation to the 2013 Hidy paper, but this insight (ie. extensive CIS far inland – in fact, the most extensive CIS advances in interior Yukon are the earliest ones) could be better incorporated in the penultimate paragraph of the introduction (lines 15-24, page 2) and likely also in section 4.3.

2-Authors' response

Thank you for your comments. We agree that our manuscript could benefit from a tighter connection between the signs of the Cordilleran glaciation in terrestrial records and our marine record at Site U1417.

3-Authors' changes in the revised manuscript

We have briefly mentioned the timing of the expansion of the Cordilleran glaciation according to the terrestrial records in Hidy et al., 2013 and Duk-Rodkin et al., 2004 (page 2, lines 15 to 17) together with other previous studies of the timing of the Cordilleran glaciation in marine records in Gulick et al., 2014 (page 2, line 18) in our introduction. We have further mentioned these terrestrial records in Section 4.3 (page 12, lines 4 to 7).

1-Comment from the Editor

Please also consider making some reference to the well established stratigraphic, geomorphic, and paleomag record for extensive inland CIS glaciation during the late Pliocene/earliest Pleistocene – you could refer to Duk-Rodkin et al 2004 (Dev. Quat. Sci 2B: 313-345) as a starting point. I think this would strengthen the connection between your relatively distal marine record and the less-well-dated direct terrestrial evidence for ice sheet extent.

2-Authors' response

Thank you for your comments. We agree that previous studies of terrestrial records might strengthen our interpretations of the behaviour of the Cordilleran Ice Sheet during the early and mid-Pliocene and late Pliocene.

3-Authors' changes in the revised manuscript

We have included the valuable insights about the Cordilleran Ice Sheet expansion in Duk-Rodkin et al. (2004) study into our early and mid-Pliocene TAR interpretations (page 10, lines 30 to 32). We have also included Duk-Rodkin et al. (2004)'s interpretations of the development of fan deltas due to coastal uplift to support our TAR interpretations during the late Pliocene (page 11, lines 12 to 14). We have also compared the dates of the oNHG and iNHG in our late Pliocene sections with the Duk-Rodkin et al. (2004) and related papers' timing of the major glaciation in Yukon (page 11, lines 10 to 12; page 12, lines 5 to7).

1-Comment from the Editor

Thank you again for submitting this very interesting and important manuscript to Climate of the Past. I look forward to seeing your revised submission.

2-Authors' response

Thank you.

We would like to thank Anonymous Referee 2 for the constructive comments provided to help us improve our manuscript. Please find below our responses to these comments and the manuscript changes.

1-Comment from Referee

In my first review I stated to major points of criticism, i.e. the presentation of the TAR and CPI indices and the confusing presentation of section 4.4. The authors addressed both points so that the clarity of the entire discussion greatly improved overall. The new section 4.4 is much clearer than before and the authors' inferences are easy to follow, now. So, I don't have further complaints here.

2-Authors' response

Thank you.

3-Authors' changes in the revised manuscript

Please note that have further altered Figure 3 to show the records after applying a 1kyr linear interpolation and 100kyr smoothing (rather than the previous approximate 100kyr smoothing directly applied) to the raw data. Due to the continuity of the data analyses, we have represented the data as continuous lines. We have further included the atmospheric $CO_2$ record from marine $\delta^{11}B$ of Martínez-Botí et al., (2015) to the atmospheric $CO_2$ record from alkenone $\delta^{13}C$ of Seki et al., (2010) (revised in Foster et al., 2017) to represent the agreement/disparity between different atmospheric $CO_2$ reconstructions across the NHG.

1-Comment from Referee

However, I still have some concerns regarding parts of their interpretation of the TAR index which partly arise from having the CPI plotted next to the TAR. At the present stage, the inferences made from the TAR during the oNHG (3-2.8 Ma) are questionable and require further clarification by discussing the different potential factors controlling the TAR in more detail (see below). This is crucial to strengthen the key conclusions that the onset of the Cordilleran Ice Sheet glaciation occurred during this time interval. I am convinced, a more comprehensive discussion around the different factors controlling the TAR will unequivocally strengthen the inferences made from this multi-proxy approach. As already mentioned in my first review, presenting a multi-proxy approach the study is a valuable and important contribution to the ongoing discussion about the glaciation chronology in the Cordillera during the Plio-Pleistocene transition. The paper should be accepted for publication as soon as this last issue regarding the TAR has been addressed.

2-Authors' response

Thank you for your comments. We agree that a further clarification of the TAR would improve the manuscript. Please see the changes we have implemented in our comment below.

1-Comment from Referee

Interpretation of TAR during the oNHG, 3-2.8 Ma (page 10, line 40-page 11, line 5):

As the authors correctly describe in the methods section, the TAR index estimates the relative abundance of long-chain n-alkanes versus the short chain homologues and is commonly used to estimate changes in the relative contributions of plant-wax lipids from higher land plants and aquatic production (page 5, lines 4-7). In the discussion of the onset of the Cordilleran Ice Sheet glaciation (3 to 2.8 Ma) the authors infer an increase in the export of plant-derived organic matter based on a decrease in the TAR-values (page 10, line 40 to page 11, line 5). In fact, this interpretation is contradictory to the general way of interpreting the TAR according to which decreasing values would imply the opposite i.e. an increase in aquatic production and/or a decrease in the contributions from higher land plants. Although I don't doubt the possibility that the export of terrigenous material including the long-chain n-alkanes may increase despite a decrease in the TAR, this interpretation needs further justification and cannot be based on the TAR alone. Concentrations or mass accumulation rates of the short-chain and long-chain n-alkanes are needed to support this statement since they allow to disentangle the individual developments of the input of short chain and long-chain n-alkanes. It may help to compare them to concentrations of other marine biomarkers, e.g. alkenones. Moreover, there are several possibilities to explain the decrease in the TAR-index and not all of them support expanding glaciation in the hinterland. Based on the current presentation of data, the authors don't have a means to rule these options out. Firstly, there could be an increase in aquatic production, which does not necessarily mean that the controls on the export of leaf-wax biomarkers change. That means it would be possible that the vegetation cover

remains as extensive as before (during the early to mid-Pliocene) which would be in conflict with the authors' inference of advancing ice cover in the region. Secondly, assuming that the

source of the long-chain n-alkanes remains the same as during the early to mid-Pliocene, the decreasing TAR may report on a decline in the export of terrigenous organic matter attesting to e.g. a reduction of the vegetation cover. Although this view would be in harmony with advancing ice masses in the hinterland it would be in conflict with the authors' idea of enhanced discharge of leaf-wax lipids from vegetation. Thirdly, the overall low CPI-values (around 1.5) point to a high degree of degradation. So, it is plausible that a large fraction of the long-chain n-alkanes may be petrogenic instead of dominantly vegetation-derived. As the authors point out, there are coal-bearing bedrocks in the region and these are characterized by low TAR (up to 2) and low CPI-values (<1) (page 10, lines 25-28 and page 12, lines 25/26).

As such, the decrease in the TAR may also indicate intensified erosion of bedrock compared to the early and mid-Pliocene. Under these circumstances, decreasing TAR would be compatible with increased export of terrigenous organic matter. This scenario would be in accordance with the idea of reduced vegetation cover and expanding ice masses in the Cordillera.

Since the inferences for the Ice Sheet Glaciation strongly depend on the way of interpreting the TAR, the different potential controls on the index need a more detailed recognition in the discussion in order to strengthen the authors' key conclusions, namely that the onset of the Cordilleran Ice Sheet Glaciation occurred between 3 and 2.8 Ma.

2-Authors' response

Thank you for your comments. We agree that the TAR during the 3 to 2.8 Ma needs clarifying. The confusing sentence (page 10, line 40 to page 11, line 5-now clarified in page 11, lines 1 to 4) arose because we intended to outline a change of TAR pattern from above the TAR average value before 3 Ma to below the TAR average value from 3 Ma onwards (into the Early Pleistocene). Thank you for drawing attention to the confusing statements.

3-Authors' changes in the revised manuscript

We have addressed this issue by explaining first the TAR during the 3-2.8 Ma (page 10, line 41 and page 11, lines 1 to 10) and then during the 2.7-1.7 Ma time period in Section 4.3 (page 12, lines 22 to 28). To help visualising the discussion behind the TAR, we have included Table 2. Table 2 includes the above or below 4-1.7 average TAR at Site U1417 and terrigenous and aquatic n-alkane mass accumulation rates during the time periods discussed in the text. TAR lowers during the 3-2.8 Ma due to a slight decrease in terrigenous organic matter and a slight increase in aquatic organic matter (page 11, lines 1 to 4). We interpret this as an increase in ice-cover and lower plant wax contribution to the mix of sources of terrigenous OM eroded and transported to Site U1417, despite the overall increase in sedimentation rates (page 11, lines 8 to 10). The increase in sedimentation rates could be related to a change in erosional pathways (i.e., to a softer bedrock) but because our CPI remain similar than during the 4-3 Ma, we interpret rather an increase in erosion due to an advancing ice sheet (page 11, lines 5 to 8). This interpretation also fits well with previously published literature (page 11, lines 12 to 14).

Detailed comments

1-Comment from Referee

Page 5, line 9: remove the sentence in line 9. This seems to be out of context here in the method section. The following paragraph encompassing lines 10-16 stands well by itself.

2-Authors' response and changes in the revised manuscript

This sentence has been modified and moved to Section 4.3 (page 12, lines 22 to 24).

1-Comment from Referee

Page 10, line 20: n-alkane => n-alkane

2-Authors' response and changes in the revised manuscript

This has been modified (page 10, line 20).

1-Comment from Referee

Page 12, line 12-13: Actually, the CPI cannot attest to the maturity of the short-chain alkanes as the formula used here does not consider the homologues shorter than C24.

2-Authors' response and changes in the revised manuscript

This has been now amended and CPI only refers to the maturity of terrigenous OM (page 12, lines 26 to 28).

1-Comment from Referee

The interpretation of the TAR should be justified a little more in detail considering the different interpretation of a similar signal for the period from 3-2.8 Ma. It is not clear why the low TAR values are interpreted as stemming from increased productivity from 2.7 Ma onwards while between 3-2.8 Ma the progressive decrease towards these low values is attributed to an increased export of leaf-wax lipids from land plants. Please, explain.

2-Authors' response and changes in the revised manuscript

This has now been amended (page 10, line 41 and page 12, lines 1 to 14). Table 2 has been included (Page 25).

1-Comment from Referee

Page 2, lines 17 and 20: In line 17 it says St. Elias Mountains while in line 20 it is St Elias Mountains. Please, unify the spelling throughout the entire manuscript.

2-Authors' response and changes in the revised manuscript

This has now been unified to read "St. Elias mountains" throughout the manuscript.

[revised manuscript text omitted]